# Statistical prediction of microbial metabolic traits from genomes

**Zeqian Li**[1,2,3], **Ahmed Selim**[4], **Seppe Kuehn**[1,2]*

**1** Center for the Physics of Evolving Systems, The University of Chicago, Chicago, Illinois, United States of America, **2** Department of Ecology and Evolution, The University of Chicago, Chicago, Illinois, United States of America, **3** Department of Physics, The University of Illinois at Urbana-Champaign, Urbana, Illinois, United States of America, **4** Graduate Program in Biophysical Sciences, The University of Chicago, Chicago, Illinois, United States of America

\* seppe.kuehn@uchicago.edu

**Data Availability Statement:** Raw sequencing data and genome assemblies are available at the original source of each study (NCBI BioProject PRJNA660495 and PRJNA513156 for genomes from Gowda et al. (2022), PRJNA540276 for genomes from Muscarella et al. (2019), PRJNA940744 for genomes from Prabhakara et al.

## Abstract

The metabolic activity of microbial communities is central to their role in biogeochemical cycles, human health, and biotechnology. Despite the abundance of sequencing data characterizing these consortia, it remains a serious challenge to predict microbial metabolic traits from sequencing data alone. Here we culture 96 bacterial isolates individually and assay their ability to grow on 10 distinct compounds as a sole carbon source. Using these data as well as two existing datasets, we show that statistical approaches can accurately predict bacterial carbon utilization traits from genomes. First, we show that classifiers trained on gene content can accurately predict bacterial carbon utilization phenotypes by encoding phylogenetic information. These models substantially outperform predictions made by constraint-based metabolic models automatically constructed from genomes. This result solidifies our current knowledge about the strong connection between phylogeny and metabolic traits. However, phylogeny-based predictions fail to predict traits for taxa that are phylogenetically distant from any strains in the training set. To overcome this we train improved models on gene presence/absence to predict carbon utilization traits from gene content. We show that models that predict carbon utilization traits from gene presence/absence can generalize to taxa that are phylogenetically distant from the training set either by exploiting biochemical information for feature selection or by having sufficiently large datasets. In the latter case, we provide evidence that a statistical approach can identify putatively mechanistic genes involved in metabolic traits. Our study demonstrates the potential power for predicting microbial phenotypes from genotypes using statistical approaches.

## Author summary

The metabolic activity of microbes is essential to sustaining life on Earth, biotechnological processes, and host fitness. As a result, the metabolic traits of microbes have been a focus of microbiology and microbial ecology for centuries, historically relying on painstaking laboratory experiments. Sequencing technologies have given us an unprecedented look at

(2023) (sequenced in this study), and Gralka et al. (2023)). The analysis data are publicly available on Open Science Framework (https://doi.org/10.17605/OSF.IO/JWKR7). The bioinformatic pipeline and all data analysis code are available at https://github.com/zeqianli/CarbonUtilization.

**Funding:** SK and ZL acknowledge funding from the National Science Foundation Biology Directorate (MCB 2117477). SK acknowledges funding support from the National Science Foundation Biology Directorate (MCB 1921439), and National Institutes of Health (1R01GM151538). All three of these grants provided salary support to SK. S.K. acknowledges support from the National Science Foundation through the Center for Living Systems (grant no. 2317138). The funders had no role in study design, data collection and analysis, decision to publish, or preparation of the manuscript.

**Competing interests:** The authors have declared that no competing interests exist.

microbial genomes, but connecting genomes to specific traits in non-model bacteria remained a huge challenge.

We demonstrate that simple statistical models, trained on modest datasets can outperform existing approaches for predicting bacterial carbon utilization traits from genomes. First, machine learning models can leverage phylogenetic information to predict which carbon sources a given strain will grow on more reliably than state-of-the-art constraint-based models. We extend this success to making predictions for phylogenetically distant taxa using knowledge of gene presence and absence alone and show that these approaches can reveal mechanistic insights into the genomic basis of bacterial traits. Our study takes steps to shift the existing paradigm from using metabolic models to using statistical methods to predict microbial traits from genomes.

This study leveraged new trait measurements on 100 sequenced bacterial isolates as well as large-scale datasets spanning over 4000 genomes curated from databases. We develop a suite of computational approaches from first principles that enable insights into when, why, and how statistical predictions succeed or fail.

## Introduction

Microbial communities play a crucial role in global nutrient cycles, human health, pharmaceutical, and technological production processes [1–4]. In these contexts, the metabolic activity of bacteria is of central importance. For example, in the gut, bacteria consume complex carbohydrates to assist host metabolism [5]; in the soils, bacteria utilize organic carbon produced by photosynthetic microbes to retain soil carbon [6]; and in the oceans, microbes decompose detritus with key implications for the global carbon cycle [7]. In each of these contexts, bacteria catabolize organic carbon to generate energy and/or biomass. Thus, a key challenge in dissecting complex communities in almost any context is understanding how bacterial taxa utilize distinct carbon sources to generate energy and biomass.

DNA sequencing technologies have made it possible to access the genomic content of microbial communities at a massive scale. As a result, we now have a picture of the genomic content of microbial communities from deep-sea sediments to the human gut, giving us a picture of which microbial taxa are present, the genes they possess, and even their expression through time [8–11]. However, it remains difficult to connect taxonomic or genomic measurements to metabolic traits, including carbon catabolic traits, and this limits our ability to interpret sequencing measurements in terms of the metabolic processes that are key to microbial impacts on human and environmental health.

The challenge of decoding metabolic traits from genomes arises from the complexity of genotype-to-phenotype relationships. Genes encode enzymes that link together complex pathways of reactions. For carbon catabolism, while central carbon metabolism is well understood, a myriad of metabolic pathways break down compounds prior to central metabolism. Not only are the genes encoding these compounds not always known [12], but correctly annotating sequenced genomes remains challenging [13]. As a result, mapping gene content to metabolic networks and ultimately metabolic traits is a major challenge.

One of the most widely used frameworks for predicting metabolic traits from genomes is constraint-based modeling (CBM). CBM simplifies the problem by making a steady-state assumption to constrain possible metabolic fluxes [14]. By combining this assumption with physiological constraints, such as molecular crowding, CBM allows for predictions of metabolic fluxes and biomass production. The approach has proven powerful for dissecting

metabolic fluxes in model organisms where extensive experimental data are available to constrain models [15, 16]. Recent advances have enabled the construction of constraint-based models directly from genomes of natural isolates, opening the door to using CBM to predict metabolic traits for non-model organisms [17]. However, predicting metabolic traits from genomes for non-model bacteria at scale remains difficult, due to the lack of experimental data and the significant manual curation required [18].

Recent studies have shown that statistical methods that utilize gene content alone can yield remarkably quantitative predictions of metabolic traits in simple metabolic pathways such as denitrification [19]. However, it is unclear if such predictions are feasible for complex metabolic pathways such as the catabolism of diverse organic carbon compounds.

Here we demonstrate the high-accuracy prediction of carbon catabolic traits for non-model bacteria from genomes alone using a statistical approach. We combine new and existing datasets measuring the aerobic growth of diverse bacterial isolates on a variety of carbon sources. We show that for most carbon sources, machine learning models leverage phylogenetic information to predict carbon utilization traits (whether a strain can grow on a carbon source or not) from genomic information. These predictions succeed when the target bacterial taxa is closely related to a strain in the training dataset, but fail when this is not the case. The strong correlation between phylogeny and microbial traits was well studied in the literature [20, 21] and has been practiced both clinically and in the lab to identifying taxa from metabolic profiling [22, 23]. Our results provide further support of this approach's high accuracy and broad applicability. However, this method breaks down when predicting traits for phylogenetically distant strains. To tackle this problem, we further developed models that utilize a biochemically informed feature selection process, to rationally choose genes to use to predict traits from genomes, can improve predictions for taxa that are distant from any member of the training dataset. In addition, when datasets are large ($\sim$1000 taxa) trained models are able to predict traits for taxa that are distant from any member of the training data. In these cases, we hypothesize that classifiers are learning genes mechanistically relevant to the predicted traits, not phylogenetic information.

Our study provides a new conceptual approach to predicting metabolic phenotypes from genomes. We propose that rather than studying metabolic networks in model organisms to construct CBMs for non-model bacteria, a purely statistical approach, with sufficiently large datasets, can be used to discover the salient genomic features that enable specific metabolic capabilities in microbes. This new approach could enable the large-scale interpretation of sequencing data in terms of the metabolic capabilities of bacteria present in microbiomes across the globe.

## Results

In this study, we aim to predict the ability of single microbial strains to utilize specific compounds as sole sources of carbon (Fig 1A). To achieve this, we collected 96 diverse bacteria isolated from natural environments drawn from three previous studies [19, 24, 25]. For each strain in this library, we measured whether they were able to grow or not on each of 10 diverse carbon sources. In these growth assays, we included a small quantity of yeast extract to satisfy potential auxotrophies (Methods). Growth was determined by measuring optical density after 72 hours of incubation at 30˚C. These experiments resulted in a 96 by 10 growth/no-growth binary matrix. All strains in our library were either previously sequenced or sequenced in this study (Fig 1B; Methods). Sequenced genomes were annotated for gene content using established methods (Methods).

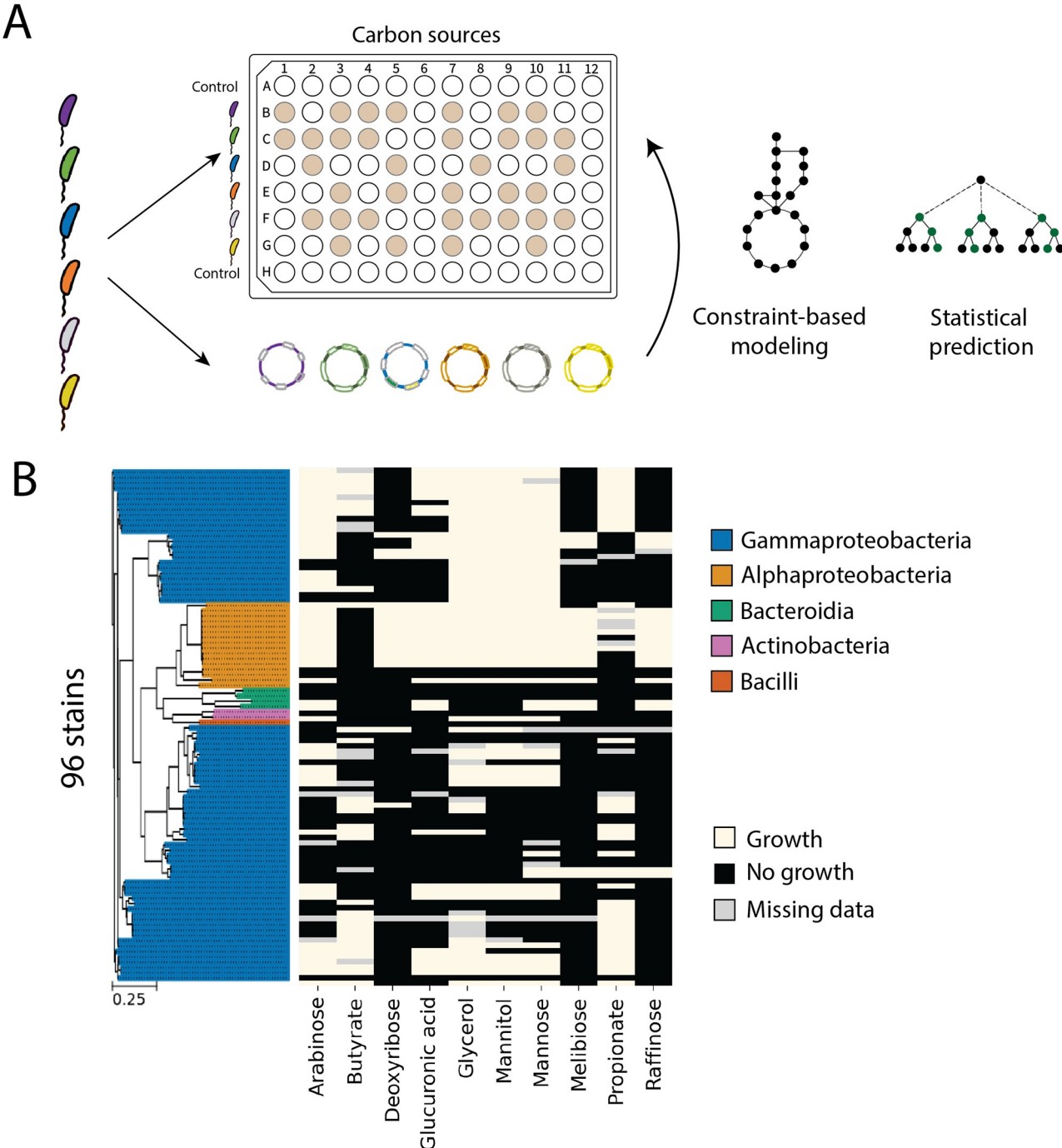

**Fig 1. Predicting microbial carbon utilization traits from gene content.** (A) Experimental setup to assess the ability of 96 bacterial strains to grow on 10 different organic carbon compounds as a sole carbon source. We employed both constraint-based modeling (CBM) and statistical models to predict the microbial carbon utilization traits (growth/no-growth) from genomes. (B) The dataset consists of a 96 x 10 binary matrix, representing the growth of the 96 strains in our library. Growth was assessed by measuring optical density (OD) after 72 hours of incubation as described in Methods. Growth/no-growth traits are indicated in the legend. The tree to the left indicates phylogenetic relationships between strains, with colors indicating classes as given in the legend. The scale bar shows the tree branch length.

## Automated constraint-based modeling fails to predict microbial carbon utilization traits

First, we created genome-scale metabolic models for our collection of strains using established automated pipelines [17], simulated their growth on media with each carbon compound used in this study as sole carbon sources, and compared the predictions with experimental data (Methods). With the exception of raffinose, we found that constraint-based models (CBM) provided inaccurate predictions, with mean accuracy that was comparable to random guesses generated by a null model (Fig 2B; Methods). This result is consistent with previous studies applying automated constraint-based modeling to non-model organisms [17, 19, 26].

Gap-filling is a common strategy to improve metabolic models, which adds potential missing genes in the genome annotation by enforcing growth in a media condition that should guarantee growth [17]. We performed gap-filling by enforcing growth on a defined minimal media which resembled the experimental media with glucose as the carbon source. We found that gap-filling did not improve model prediction accuracy (S2(A) Fig), and instead shifted the model behavior from overly specific to overly sensitive (S2(C) Fig), i.e, the prediction changed from having a high false-negative rate to a high false-positive rate. Changing the carbon source in the gap-filling media did not alter prediction accuracy (S2(B) Fig). We also attempted to enforce metabolite uptake to account for potential missing transporters in the genome annotation, which did not improve predictions (S2(A) Fig).

It is worth noting that, on melibiose and raffinose, two polysaccharides that can be degraded to simple sugars (glucose and fructose) through one or two reactions [27, 28], CBM made more accurate predictions (Fig 2B). On carbon sources with complex metabolism to be utilized as sole energy sources, such as organic acids (butyrate and propionate), CBM prediction accuracy was very low. Therefore, we hypothesize that CBM prediction performance is metabolite-specific and that compounds with simpler catabolic pathways are easier to predict.

## Machine learning accurately predicts carbon utilization traits using genes by exploiting phylogeny

Given the challenges of predicting traits for non-model bacterial strains using constraint-based modeling derived directly from genomes, we sought an alternative approach. Previous studies showed that, in simpler metabolic pathways, statistical models using gene presence/absence as predictors can accurately predict metabolic traits [19]. Given the previous success, we asked whether similar statistical methods apply to the more complex carbon metabolic processes.

To proceed, we annotated the genomes of the 96 strain in our library with KEGG ortholog groups (KO) [29] and created a 96 × 6746 KO presence-absence matrix (Methods). We asked whether random forest (RF) classifiers trained using KO presence-absence as predictors could predict carbon utilization traits. To evaluate the predictions, the data were randomly partitioned into training (80% of the samples) and test sets (20% of the samples, Fig 2A). The RF model demonstrated high accuracy in predicting carbon utilization and consistently outperformed the null models across all carbon sources (Fig 2B), achieving an accuracy of over 90% in most cases. As a result, random forest models based on gene presence/absence substantially outperformed CBM predictions on all 10 carbon sources in our dataset.

It is important to note that the prediction accuracy is a biased metric and is strongly influenced by the frequency of growth (1) and no-growth (0) entries in the trait data. To account for this bias, we always compare the distribution of prediction accuracy scores over data partitions with two null models: Bernoulli null, which randomly guessed growth based on the empirical probability of growth on the training set ($p$); and identity null, which always

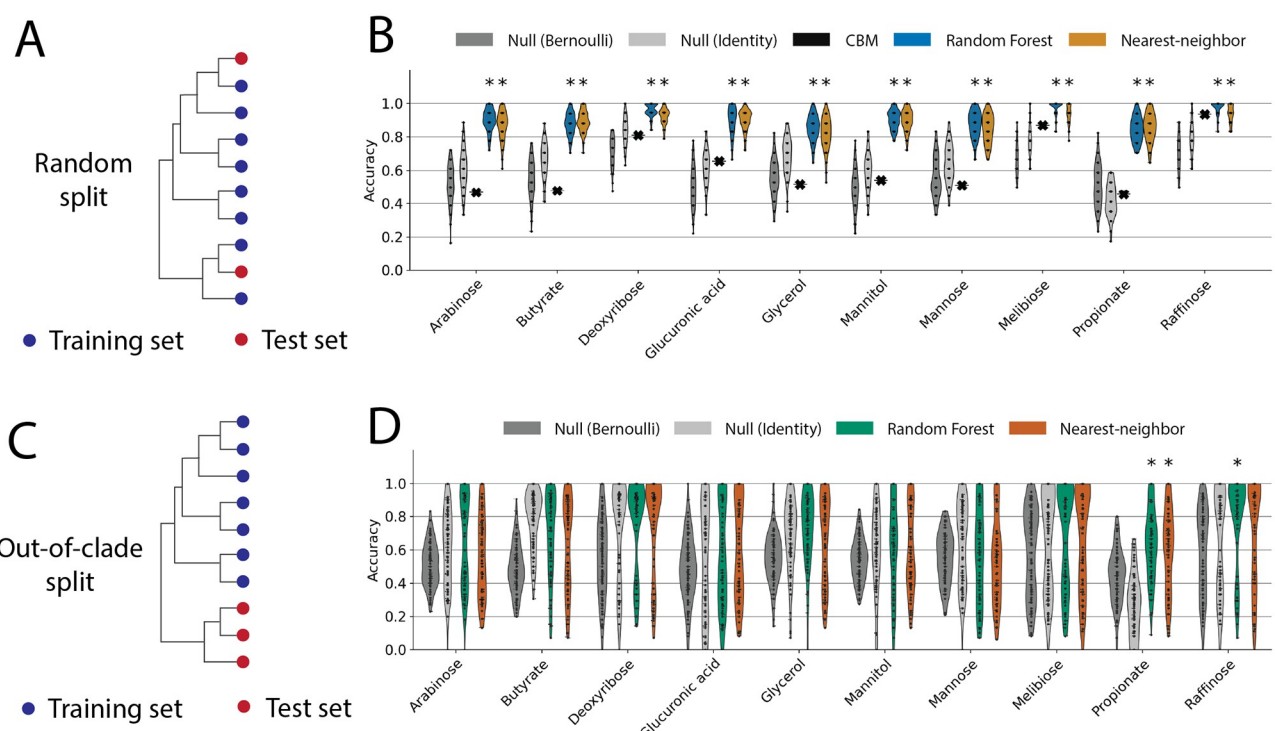

**Fig 2. Carbon utilization trait prediction accuracies using constraint-based modeling (CBM) and machine learning.** (A) Schematic of randomly selected test and training sets. Strains (colored circles) on the phylogenetic tree are chosen at random, blue for training and red for testing. (B) Trait prediction accuracies using constraint-based models generated from genomes (black markers), comparing with random forest classifiers (blue) and nearest-neighbor classifiers (orange) trained using KO presence-absence as predictors with randomly partitioned training/test sets as in (A). Predictions are tested against two null models (Bernoulli, identity; see Methods). Models that significantly outperform both null models are marked * above ($p < 0.05$ after multiple-testing correction; see Methods). (C) Schematic of out-of-clade train/test partitions. Note the test set (red) comprises a group of phylogenetically related strains (clade). (D) Trait prediction accuracy for out-of-clade test sets (as in C), using random forest classifiers (green) or nearest neighbor classifiers with KO presence-absence. Predictions that significantly outperform both null models are marked * at the top ($p < 0.05$).

predicted the most frequent trait (growth or no growth) from the training set. The two null models represent two kinds of uninformed predictions, which sometimes make seemingly accurate predictions as a result of unbalanced traits with high frequencies of either 0 or 1 entries. A model is considered successful if it has an accuracy distribution, across random partitions of the data, significantly higher than both null models (Methods). In the case of random forest classifiers, the predictions were significantly higher when tested against both null models (Fig 2B; $p < 0.05$ after multiple-testing correction; Methods).

We next wanted to explore the underlying mechanisms that enabled these accurate predictions. Specifically, we computed the mean feature importance of KOs across the random forest models trained on different data partitions. We wondered whether the RF model was using the presence and absence of genes relevant to the carbon utilization trait being predicted. Surprisingly, we found that the RF classifiers often utilized genes that were not directly involved in carbon source utilization (S3 Fig).

If the trained models encode meaningful biochemical processes such as key carbon catabolic enzymes, as has been observed in previous studies [19], then we would expect these biochemical processes to be universal, allowing the models trained on one dataset to be predictive on phylogenetically distant strains. To test this hypothesis, we evaluated the prediction model using a different cross-validation approach, which we refer to as the "out-of-clade" partition.

Briefly, we constructed a phylogenetic tree of the 96 strains using their 16S rRNA sequences and selected clades of closely related strains to reserve for testing, while training the model on the remaining strains (Fig 2C; Methods).

We found random forests were less accurate and did not significantly outperform both null models under out-of-clade cross-validation (except growth on propionate, Fig 2D) and varying hyperparameters did not improve predictions (S4 Fig). We also found similarly poor predictions using other classification techniques (S6(C) Fig; Methods). Our inability to predict traits "out-of-clade" suggests that in our original classification, the RF models were exploiting phylogenetic correlations [30] to make predictions of traits. To test this idea, we asked how well phylogeny alone predicts whether or not a bacterium could grow on a carbon source.

We used a simple nearest-neighbor classifier [31] using 16S rRNA sequences as predictors, which predicted the trait of a target strain by assigning it the same trait as the most closely related strain in the training set (Methods). The nearest neighbor model accurately predicted carbon utilization and significantly outperformed both null models on all carbon sources, except deoxyribose (S5(A) Fig), with an accuracy of approximately 90% in most cases. However, nearest neighbor predictions do not work out-of-clade (S5(B) Fig) as we expect. This is because phylogenetically close strains are more likely to exhibit the same trait up to a certain evolutionary distance (S5(C) Fig). [20]. We calculated the phylogeny-trait correlation length scale for all carbon sources in the dataset (Methods; S5(D) Fig). We found that for predictions that succeed (random partition), the phylogenetic distance between strains in the test set and the training set is smaller than the phylogenetic distance over which traits are correlated. Conversely, out-of-clade predictions fail because test set strains are frequently phylogenetically far away from any member of the training set.

This result strongly supports the idea that our original classification (Fig 2B) exploits phylogenetic information in gene content to predict traits. We validated the notion that random forest classifiers exploit phylogenetic information by showing that, under both random and out-of-clade partitions, random forest classifiers tend to make the same prediction for strains with similar genomes (defined by L1 distance of KO presence-absence), consequently behaving like nearest-neighbor classifiers based on genome similarity (Methods; S6 Fig). Because genome similarity strongly correlates with phylogeny (S6(A) Fig), random forest classifiers trained on gene presence-absence showed similar behavior as classifiers using 16S sequences.

Next, we asked whether or not the results we obtained for the strains in our library held more broadly for other bacterial strains and carbon sources. To do this, we repeated the analysis on an existing dataset curated from [26], with growth data of 172 diverse marine bacteria on 100 carbon sources including various sugars, sugar alcohols, organic acids, and amino acids (Methods; Fig 3A). Note that some strains and carbon sources from [26] are excluded from the analysis due to annotation difficulty or lack of variation across strains in the growth/no-growth trait which inhibited model training (Methods).

We observed similar results that phylogeny was a strong predictor for growth on most carbon sources. We performed nearest neighbor classification independently on each of the 100 carbon sources assayed using only 16S rRNA sequences as features (S10(A) Fig). If the data were partitioned randomly into training and test sets, we found that 76 out of the 100 carbon sources were predicted better than both null models (S10(A) Fig). In contrast, when we tested these predictions using our out-of-clade hold-out method we found that for the vast majority of carbon sources (87 out of 100), our predictions are no better than either null model (S10(A) Fig). Consistent with this observation and our results above, when we measure the phylogenetic correlation of carbon utilization traits in this dataset, we see a similar level of phylogeny-trait correlation to what we observed in our smaller dataset (compare S5, S10B and S10C Figs). Training nearest-neighbor classifiers using KO presence-absence vectors as features, another

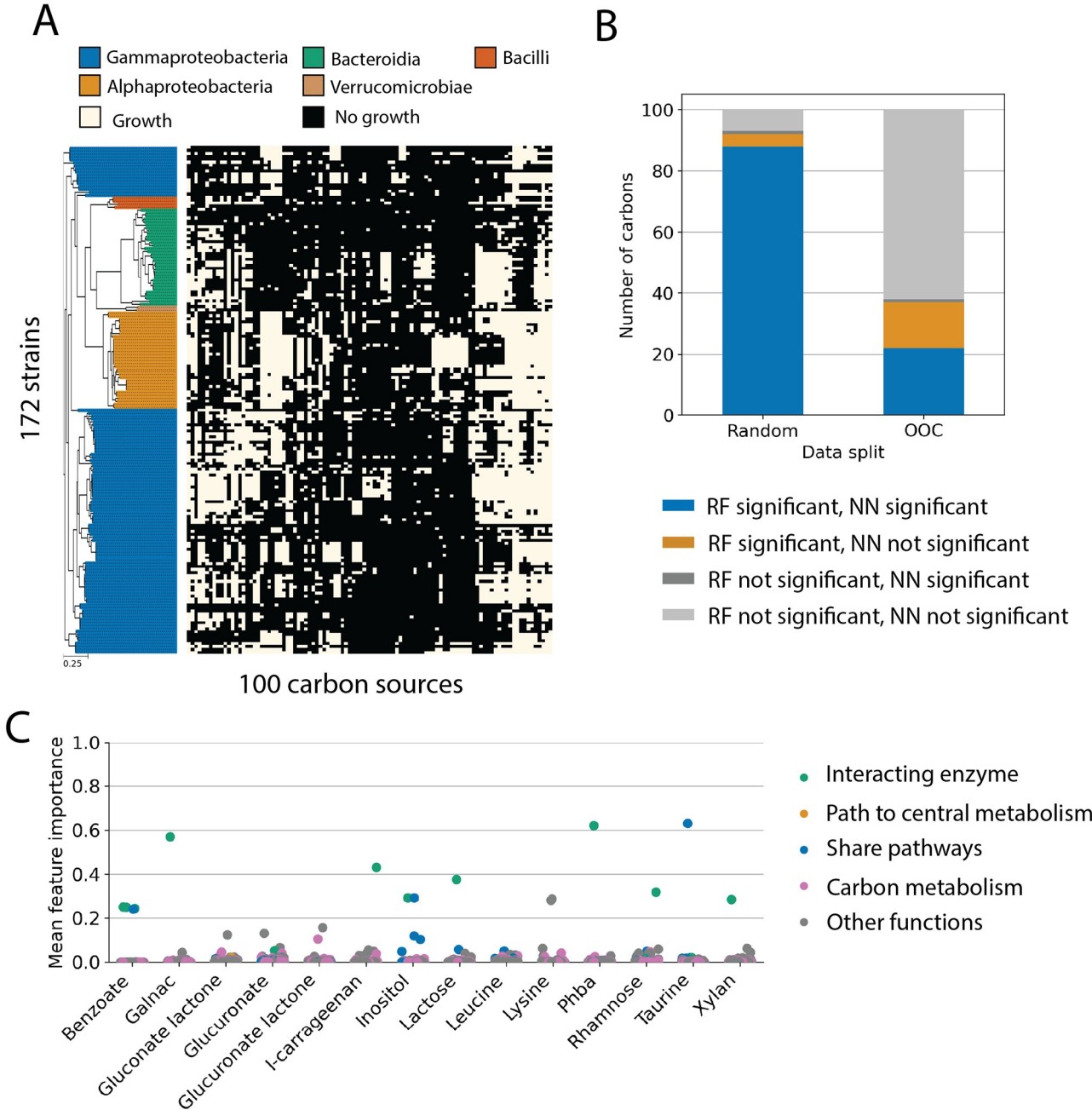

**Fig 3. Machine learning prediction of carbon utilization data from Gralka *et al.* [26].** (A) The curated dataset consists of binary growth data of 172 diverse marine bacterial strains on 100 carbon sources. Data is presented identically to Fig 1B. (B) The number of carbon sources with significant prediction accuracies compared to the two null models (Methods) by random forest or nearest neighbor classifiers under random (left) or out-of-clade (right) data partitions. Under random data partitions (left), the random forest classifier significantly outperformed both null models for 88 out of 100 carbon sources (blue and yellow). Using out-of-clade data partitions, the random forest model outperformed both null models for only 36 out of 100 carbon sources. Out of the 36 carbon sources, 14 (yellow) were not significantly predicted by the nearest-neighbor classifier, indicating that the success of random forest classifiers could not simply be attributed to phylogeny. (C) Gene feature importance scores for the 14 carbon sources where the random forest model outperformed the nearest neighbor classifier (orange in panel B). The feature importance score was calculated as the mean of feature importance scores for 100 random forests trained in the 100 out-of-clade data partitions. Gene functions are highlighted by color and were assigned using KEGG (Methods). Note that among these successful predictions by random forest, most relied on enzymes closely related to the catabolism of the carbon source (green; note the high feature importance score).

good representation for phylogeny, yielded similar results (significant predictions on 84 out of 100 carbon sources on the random partition and 24 out of 100 carbon sources on the out-of-clade partition; Fig 3B). We note that in this scenario, nearest-neighbor prediction using KO presence-absence vectors yielded better results than 16S sequences. We hypothesize that this is because distances between KO presence-absence vectors may encompass more information than phylogenetic similarities, such as metabolic preferences [26].

The high accuracy of phylogeny-based predictions of carbon utilization traits suggests that one strategy for predicting the carbon utilization ability of an uncharacterized microbial strain from a natural environment is to build a large database of phenotyped isolates. Training classifiers on these data could allow the generalizable predictions of traits from amplicon sequencing data alone. It is important to recognize that the correlation between phylogeny and metabolic traits is known and has been recognized clinically and in the lab as an approach to identifying taxa from metabolic profiling [22, 23]. Our results provide further quantitative evidence of the high accuracy and broad applicability of such predictions. This method can be useful in applications such as the human gut microbiome, where large libraries of isolates are available [32]. However, exploiting phylogeny to make predictions will, in general, not provide mechanistic insights into which genes are important for the trait being predicted.

## Predicting traits from gene presence-absence can generalize out-of-clade

We next asked whether any of the classifiers we trained using gene presence/absence to predict traits might be utilizing mechanistic information to generalize successfully out-of-clade.

For random forests classifiers trained on the data of Gralka *et al.* [26], for some carbon sources, we found that the RF classifier made better predictions than both null models on 36 out of 100 carbon sources when partitioned out-of-clade (Fig 3B). Of these 36 carbon sources, 14 yielded predictions better than the null models by the RF classifier trained on gene presence/absence and were not well-predicted by a nearest-neighbor classifier trained on gene presence-absence distances out-of-clade (Fig 3B). This observation suggests that for these 14 carbon sources, the RF models are exploiting information beyond phylogeny to make predictions from gene presence/absence. Using feature importance scores of the RF classifiers on these 14 carbon sources, we found that RF made predictions using key metabolic enzymes relevant to the catabolism of these carbon sources (Fig 3C). For instance, RF predicted benzoate utilization traits out-of-clade with 100% accuracy, and the success was almost completely attributed to 4 genes, the *benABCD* gene cluster which encodes enzymes that degrade benzoate to catechol [33]. Fig 3C highlights the functional roles of the key predictors for these 14 carbon sources.

We hypothesize that the reason for the success of out-of-clade prediction by random forests is twofold. First, we found that these 14 carbon sources consisted of amino acids and sugar with one or a few well-characterized degradation pathways. Second, the phylogeny-trait correlation length scales for these 14 carbon sources are shorter than the rest of the carbon sources ($p = 0.03$, permutation t-test; S10(C) Fig). As a result, during model training, the model can utilize higher variation, and weaker phylogenetic correlation, in the trait space to find mechanistic features. Therefore, we hypothesize that the random forest classifier is capable of making good predictions on carbon sources with well-characterized and well-annotated metabolic pathways and weaker trait correlation with phylogeny.

However, machine learning models are still not able to make informed predictions of carbon utilization traits for most carbon sources, many of which are key metabolites in shaping microbial activities in natural environments [5]. As discussed above, given the success of NN classifiers based on 16S information alone we believe that one of the challenges of using gene

presence/absence to predict traits is that there is a strong phylogenetic correlation in gene content with traits. Specifically, closely related strains have similar gene presence/absence patterns and any classifier can simply exploit this correlation to use phylogenetic information to make predictions (see Methods for a detailed discussion). Indeed, this is consistent with the observation in Figs 2D and 3B where RF classifiers' success drops significantly for out-of-clade predictions.

To overcome this strong phylogenetic effect and learn mechanistically informative and generalizable predictions from gene presence/absence, we propose two strategies: (1) feature selection, and (2) increasing sample size. We now discuss these two approaches in turn.

## Biochemically informed feature selection improves predictions of carbon utilization traits from gene presence/absence alone

We reasoned that restricting the number of features (genes) during training RF models may avoid exploiting phylogeny and enable out-of-clade predictions to succeed. First, we attempted statistical methods for feature selection. We implemented a bottom-up feature selection approach where we iteratively selected genes with the lowest conditional entropy with the carbon utilization traits in the test set (Methods). However, we found that feature selection did not improve model predictions out-of-clade (S7 Fig). The failure of this approach arises from the fact that there are a large number of features and a small number of samples. As a result, there are many sets of genes with the *equivalent* predictive power of the traits in the training set, yet not all choices are equally predictive of traits on the test set. So, there is no obvious way to choose a good feature set (to predict the traits of those held-out strains), resulting in poor prediction performance (Methods).

We then attempted to select particularly generalizable features by implementing meta-learning, a common approach in the field of domain generalization [34] (Methods). To implement meta-learning, we created meta-partitions from the training data following the out-of-clade regime, where we selected features that enabled accurate out-of-clade predictions in the meta-partitions, and then testing the selected features on the true held-out clades (test set). However, we did not observe improvement in prediction accuracy (S7 Fig), likely due to the same reason of redundant feature sets providing equivalent predictive power across the training sets (Methods).

The failure of feature selection based on statistics indicates that additional information is needed to distinguish predictive features. One way to do so is to select genes associated with the catabolism of particular compounds. Although identifying the exact set of such genes and encoded enzymes can prove challenging due to ambiguities of annotation [13] or enzyme promiscuity [17], we can still restrict the set of genes based on biochemical information.

To accomplish this, we employed KEGG pathways [29, 35, 36] to select genes relevant to the utilization of each of the 10 carbon sources in our dataset and trained statistical models using these genes as predictors. We took a hierarchical approach to select genes within KEGG pathways: (1) all genes present in the KEGG pathways related to the target carbon source, (2) genes on the metabolic paths from the carbon source to the central carbon metabolism, and (3) genes encoding enzymes that catalyze reactions involving the carbon source (Fig 4A; see Methods).

Restricting genes to biochemically relevant ones resulted in improved out-of-clade predictions for multiple carbon sources (arabinose, butyrate, glucuronic acid, mannitol, mannose, and raffinose). We saw the greatest improvement in predicting growth on arabinose (compare Figs 2D and 4B), where RF trained on selected KOs on the biochemical paths to the central metabolism reaches a mean prediction accuracy over >90%, outperforming both null models

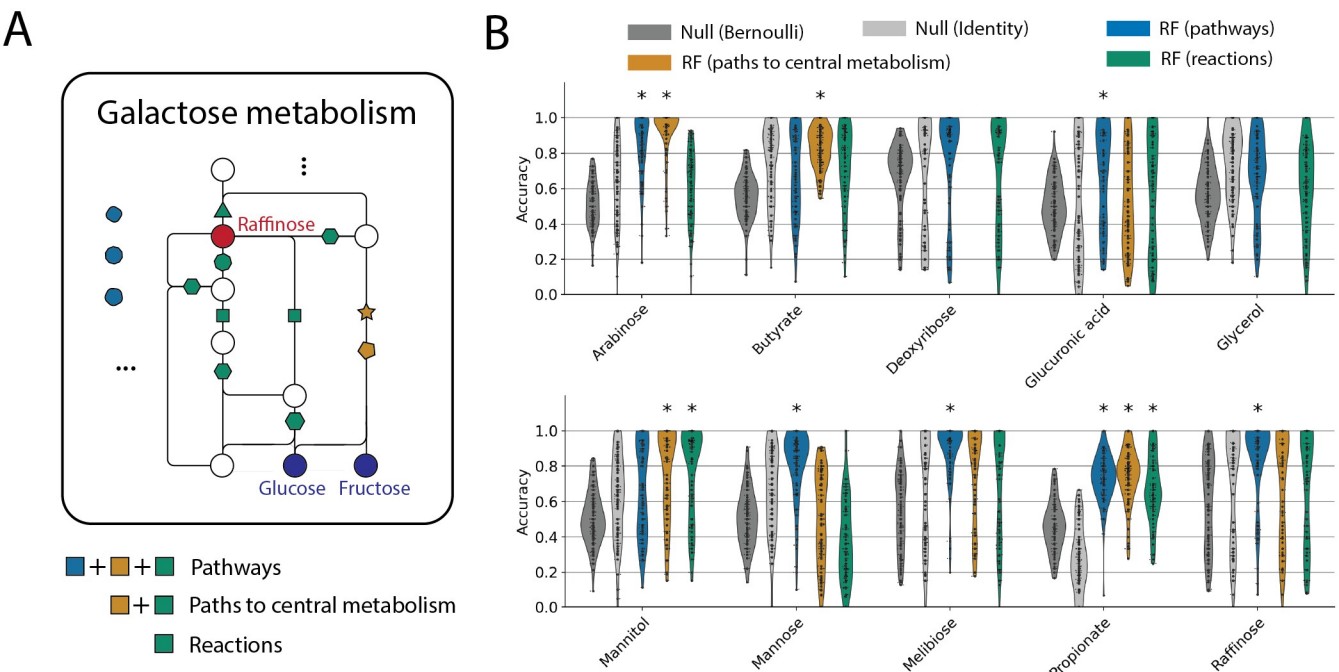

**Fig 4. Selecting genes for predictions using biochemical information improves out-of-clade predictions.** (A) Schematic of hierarchically selecting genes (KOs) for prediction using biochemical information. The panel shows part of the KEGG galactose metabolism pathway, the central metabolic pathway for raffinose catabolism in the KEGG pathway database [29, 35, 36]. Circles indicate metabolites, lines indicate reactions and colored polygons indicate KOs responsible for reactions. Polygons of the same shape are annotated as the same KO that can perform more than one reaction. The metabolic network shown includes only a subset of the reactions in the pathway and omitted reactions and their KOs are indicated by the polygons on the left. Given a carbon source, we select KOs as predictors for random forest using three levels of specificity: all KOs in the pathways related to the carbon source (blue, yellow, and green), KOs that catalyze reactions involved the top 3 shortest paths to central carbon metabolism (green and yellow; see Methods), and KOs involved in reactions that include the supplied carbon source as a substrate or a product (green). (B) Random forest out-of-clade prediction accuracies with the three different levels of feature selection. Classifications that significantly outperformed both null models are marked with * ($p < 0.05$ with multiple testing corrections, Methods).

(Fig 4B). In addition, this model was interpretable: the two features with the highest feature importance, *Ab-araC* and *Ec-araA*, encode L-arabonate dehydrase and L-arabinose isomerase respectively, two key enzymes in the two bacterial arabinose degradation pathways [37, 38]. Further, the model generalized across datasets: using the model trained on the dataset from this study predicted arabinose utilization on strains from [26] with 95% accuracy (with F-score of 0.8) and significantly better than both null models (Bernoulli null accuracy 42%, identity null accuracy 9%).

On the dataset curated from [26], biochemically informed feature selection also improved out-of-clade prediction accuracy on multiple carbon sources. Again, for the 100 carbon sources, we selected relevant KOs for each carbon source using three methods and trained and evaluated random forest classifiers on the selected KOs using the out-of-clade test sets. On 11 of 100 carbon sources, compared to RF trained on all genes, feature selection prior to RF training significantly improved prediction accuracy via at least one of the feature selection regimes. On some carbon sources, this was achieved by forcing the RF to utilize key metabolic enzymes. For instance, when predicting whether a strain can utilize isoleucine, RF without feature selection was unable to identify the key enzyme leucine dehydrogenase, while after feature selection, it became the most important predictor (90.3% feature importance) and the RF prediction accuracy out-of-clade was improved (S10(D) Fig).

Clearly, biochemically informed feature selection can substantially improve trait predictions from genomes via a statistical approach. However, out-of-clade prediction of the utilization of some carbon sources remains a challenge. So far, we have been operating in a regime where the number of features (> 5000 genes) is significantly larger than the number of samples (100 − 200). Thus, another way to improve predictions is to increase the number of samples.

## Increasing sample size predicts tryptophan utilization providing putative mechanisms

We asked whether large datasets with genotype and phenotype data improve the prediction of carbon utilization traits from gene content. To accomplish this we used the BacDive database, a public collection of metabolite utilization data compiled from both the literature and experiments [39] (S11(A) Fig). We collected bacterial strains on the BacDive database with both whole genome and carbon utilization data (Methods). The dataset consisted of carbon utilization information for 4397 sequenced strains and 58 carbon sources. Since the dataset is compiled from various sources, the number of entries available for each carbon source is variable, ranging from 104 to 2394 strains.

Using this dataset we first tested whether phylogeny was a strong predictor of carbon utilization traits. Surprisingly, we found that phylogenetic information alone did not accurately predict traits for most carbon sources on random partitions (52 out of 58 carbon sources) and failed out-of-clade predicts for all carbon sources (S11(B) Fig). We discovered that this failure arose from the fact that, for most carbon sources, strains with growth data are phylogenetically distant. The phylogeny-trait correlation length for a trait was either not detected (S11(D) Fig) or smaller than the average phylogenetic distance between train and test strains (S11(E) Fig). As a result, phylogeny-based prediction (nearest-neighbor classifier using phylogenetic distances) failed (S11(B) Fig).

For out-of-clade predictions, RF models trained on gene presence/absence outperformed both null models on 2 carbon sources, neither of which could be predicted by NN classifiers. One of these carbon sources was tryptophan (Fig 5A), the carbon source with the largest sample size (2394). For the out-of-clade test sets, random forest models using all KOs as independent variables predicted whether or not a strain could grow on tryptophan with a mean accuracy of 92.2%. Interestingly, applying the aforementioned feature selection (Fig 4A) marginally improved the RF prediction accuracy to 93.5% by using only using genes encoding enzymes that catalyze tryptophan reactions.

On both RF models (with or without feature selection), we found the gene with the highest feature importance was *tnaA* (Fig 5B), which encodes tryptophanase, an enzyme commonly used by bacteria for tryptophan utilization [40]. This result confirms classifiers can learn mechanistically relevant genes for metabolic traits. However, the prediction using *tnaA* alone is significantly less accurate (Fig 5A, green), than models that contain secondary genes for prediction, and the two models differ in those genes used in addition to *tnaA* to predict tryptophan utilization.

Without feature selection, the gene with the second highest feature importance in trained random forests was *rnfG* (Fig 5B), which encodes a subunit of the Rnf complex, a membrane $Na^+$ pump that oxidizes reduced ferredoxin [41]. While the effects of the Rnf complex on bacterial growth are largely unknown, it is thought to be crucial for energy conservation [41]. Although we found no existing studies linking the Rnf complex with bacterial tryptophan utilization, one study suggested strong $Na^+$ dependence in bacterial tryptophan transportation to utilize it as a sole carbon source in *Escherichia coli* [42]. We found among the BacDive strain, out-of-clade prediction accuracies were significantly improved for 188 strains by using all KOs

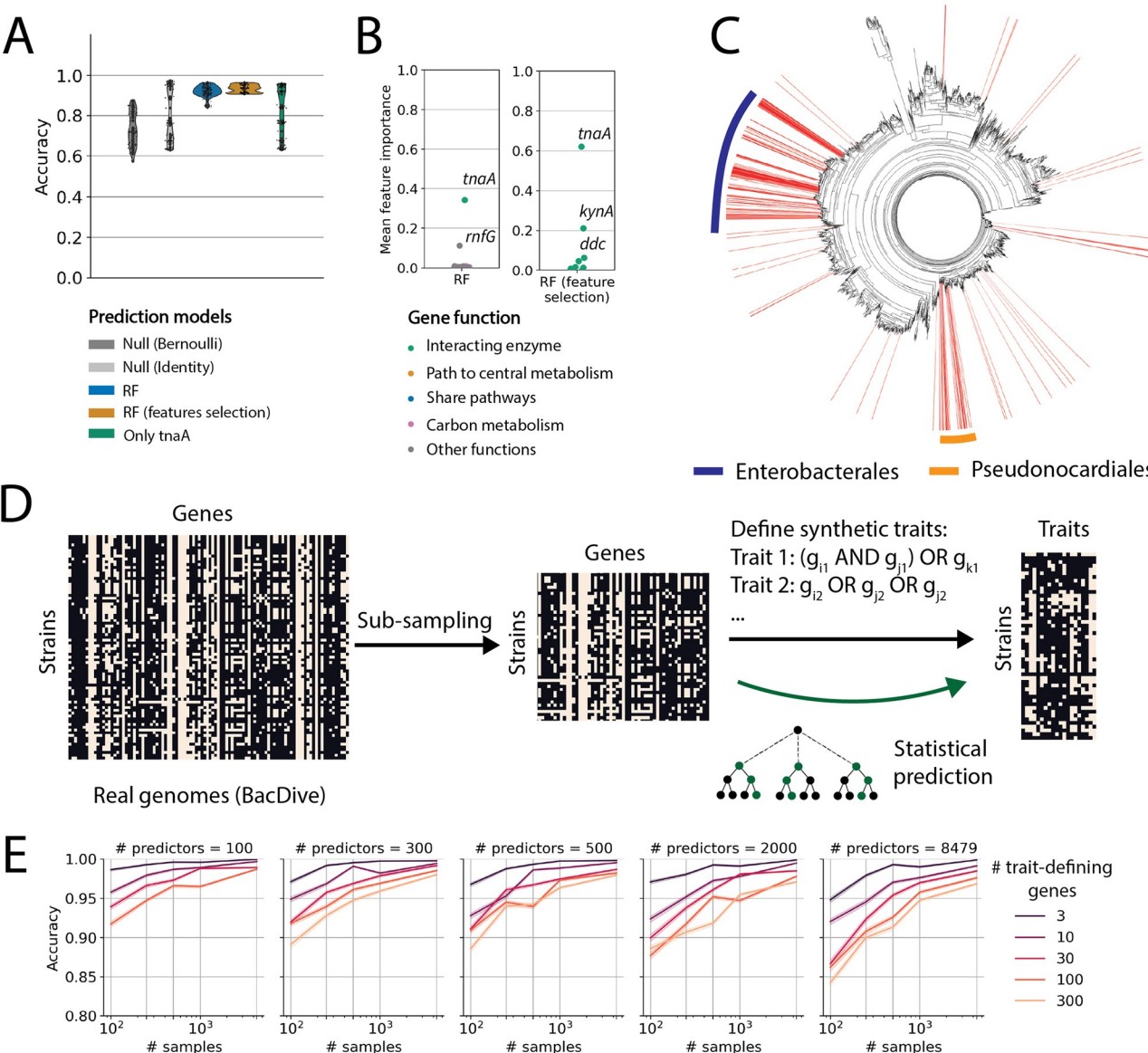

**Fig 5. Large datasets enable accurate predictions of metabolic traits out-of-clade.** (A) Out-of-clade prediction accuracy for random forest classifiers predicting growth on tryptophan from gene (KO) presence/absence for the BacDive dataset (blue, Methods) and from selected genes (orange; Methods), compared to the two null models (grey). Green shows prediction using the *tnaA* gene as the only predictor. (B) Mean feature importance scores for genes used to predict tryptophan utilization via random forest classifier using all genes (left panel, blue in C) or only genes interacting with tryptophan (right panel, yellow in C). The mean feature importance for each gene is averaged across all RF models trained on the 100 out-of-clade data partitions. Names of key features with high feature importance scores are highlighted. (C) The phylogenetic tree of the BacDive strains used for the predictions in (A). Prediction accuracies were significantly improved for 188 strains (red lines) for a random forest model using all KOs (blue in A) relative to a model using only *tnaA* (green in A). The majority of the strains (79%) belong to orders *Enterobacterales* (navy bar) and *Pseudonocardiales* (orange bar). (D) Schematic for generating synthetic traits from real genomes. The KO presence-absence matrix for real genomes is subsampled to vary the number of samples (genomes: 100, 250, 500, 1000, 4349) and the number of predictors (KOs: 100, 300, 500, 2000, 8479). Synthetic 'traits' are constructed using logical operations on columns (genes) using a variable number of trait-defining genes (3, 10, 30, 100, 300). Statistical models are assessed for their ability to predict synthetic traits from genomes. (E) Random forest prediction accuracies for out-of-clade test sets on different synthetic datasets. Synthetic datasets have three different parameters: the number of strains in the dataset (x-axis), the total number of genes used for the classification (panels), and the number of those genes used to define the trait (colors). Each dataset consists of 50 different randomly generated traits. Prediction models are evaluated over 100 out-of-clade test sets for each trait. The lines show the mean prediction accuracy across the 50 traits and the shaded region indicates the 95% confidence interval.

in the prediction compared to using only *tnaA* (Fig 5C). The majority of the strains (79%) belong to orders *Enterobacterales* (navy bar) and *Pseudonocardiales*, consistent with the previous finding in *Escherichia coli* [42]. Thus, the RF model using all genes as predictors identified a putatively important gene for tryptophan utilization. We note that in this prediction, *rnfG* did not only serve as a phylogenetic marker because the RF prediction was successful out-of-clade. We argue that the model learned a putative generalizable mechanism, and the fact that including *rnfG* improved the prediction in certain orders implies that the putative mechanism may only exist in these orders.

The random forest trained using only genes (KOs) interacting with tryptophan used a different prediction strategy and achieved marginally better prediction accuracy (Fig 5A). Similarly, *tnaA* had the highest features importance, and the features with the second and third highest feature importance scores were tryptophan 2,3-dioxygenase *kynA*, which enables the first step in an alternative tryptophan degradation pathway that converts tryptophan to kynurenic acid, and L-tryptophan decarboxylase (*ddc*), which encode another tryptophan pathway to convert it to tryptamine. Interestingly, this RF model predicts that strains can utilize tryptophan as the sole carbon source only if it possesses *tnaA* and neither *kynA* nor *ddc*. In other words, the indole pathway must be the only pathway present in a bacterial genome for it to utilize tryptophan as an energy source. Here again, the statistical approach proposes mechanistic insights that could be used to guide experiments on mutants.

## Simulations suggest large samples sizes enable out-of-clade predictions

The success of machine learning models to predict tryptophan utilization out-of-clade suggests that acquiring trait measurements for $> 1000$ strains could enable broadly generalizable predictions. We wanted to systematically investigate the effect of sample size on model predictive power. However, because different carbon sources have very different numbers of samples in the BacDive dataset, and different carbon sources are catabolized by pathways of varying complexity, studying how sample size impacts predictions systematically using existing datasets is challenging. Therefore, we chose to investigate the impact of sample size and feature number using a synthetic dataset.

To accomplish this, we defined artificial traits using logical operations on gene presence-absence and generated synthetic trait data using a large collection of real genomes (Fig 5D). Using real genomes is critical because it allowed us to examine traits while retaining the true correlations in gene presence-absence. Using this approach we computationally varied both the sample size and number of genes in each genome. We also varied the complexity of the trait by changing the number of genes used to construct the synthetic trait (Fig 5D).

We found three trends in random forest classifiers' prediction on synthetic datasets. First, the results recapitulated the patterns we saw on empirical datasets: under random partitions, random forest classifiers performed well (S12(A) Fig), significantly better than using out-of-clade partitions (compare to Fig 5E). Second, as the sample size increased, the out-of-clade prediction accuracy drastically improved. Third, more complex synthetic traits (depending on more genes) were generically more challenging to predict than simpler ones (Figs 5E and S12). The trend remains the same when the synthetic traits are modular (Methods; S13 Fig). One difference between the synthetic data and empirical data was the out-of-clade prediction accuracy for RF models for a similar number of samples (100) and predictors (8479). In this regime, predictions of synthetic traits were generally better, a discrepancy that may be due to the uncertainty of experimental data or the simplified boolean logic used to create synthetic traits.

Nonetheless, our results indicate that the accuracy of random forest out-of-clade prediction increases substantially with sample size (Fig 5E), regardless of the complexity of the trait. For a

dataset with over 1000 samples, random forests can predict all traits accurately. Moreover, consistent with our efforts to perform feature selection, we found that having fewer predictors and simpler traits defined by fewer genes can increase out-of-clade prediction accuracy.

### Interpreting predictions of traits from gene content: A case study

In our successful out-of-clade predictions of carbon utilization (e.g., tryptophan (Fig 5A and 5B), arabinose (Fig 4B)), RF models trained on gene content appear to exploit mechanistically-relevant genes to enable out-of-clade predictions. However, especially in the case of tryptophan, we do not have independent verification that the genes with high importance scores encode enzymes that are important for the utilization of the target carbon source. It could be that these genes are simply strongly correlated with the presence or absence of mechanistically relevant genes. To investigate this possibility further, we studied denitrification as a model trait where gene presence and absence are known to strongly predict phenotype and the mechanistic details of the metabolic process are well-understood [43] (Methods).

A previous study [19] characterized the denitrification phenotypes of 54 strains in our strain library. Among them, 37 strains can reduce nitrite to nitric oxide under anaerobic conditions using reductases in the denitrification pathway, and 17 strains cannot (S14 Fig). We tested whether the statistical models employed in this study could accurately predict whether or not each strain can reduce nitrite. Interestingly, we found that random forests trained using all KOs as features predict nitrite reduction almost perfectly under both random and out-of-clade prediction scenarios (S14(B) and S14(C) Fig)). However, instead of using the two enzymes (*nirS* and *nirK*) responsible for the predicted trait, the RF model utilized a gene responsible for nitric oxide reduction (*qNor*), which is a subsequent step in denitrification after nitrite reduction (S14(D) Fig). This is because nitrite-reducing bacteria, which produce nitric oxide, nearly always reduce nitric oxide as well due to its toxicity [44]. Therefore, the presence/absence of nitric oxide reductase (*qNor*) is strongly correlated with nitrite reductases (*nirS* and *nirK*), making it an excellent predictor of the target trait.

The denitrification example shows that a successful statistical model does not necessarily detect genes directly responsible for the metabolic trait of interest but exploits features correlated with the correct genes. Nonetheless, gene co-occurrence often enables the discovery of mechanistically linked genes [45], as in the case of denitrification. As a result, statistical models can guide a search for genes encoding enzymes that are mechanistically relevant for predicting traits of interest.

### Discussion

Sequencing enables us to peer into the genomic content of microbial communities at an unprecedented scale, but relating these measurements to traits presents a major challenge. Culturing microbes from communities remains challenging [46, 47]. In these situations, predicting traits based solely on sequencing data is a crucial first step in understanding and predicting microbial interactions and ultimately functions of the entire consortium.

Our work demonstrates that simple machine learning models provide a route to predicting bacterial metabolic traits. In particular, we show that whether or not a given bacterial taxon can utilize a given carbon source can be predicted from genomic information alone for some carbon sources. For some simple metabolic pathways such as melibiose and raffinose degradation, constraint-based modeling makes sound predictions. For most carbon sources, machine learning models can utilize phylogenetic signals to be a strong predictor. The results suggest that building a large reference library of diverse strains with adequate coverage on the phylogenetic tree would allow us to make predictions of metabolic traits for new, but related taxa.

While the correlation between phylogeny and traits is well known and has been applied to assign taxonomy of unknown microbes based on metabolic profiles in clinical practices and research [22, 23], results from this study validated the high accuracy of this approach in single carbon sources and its broad applicability on the bacterial tree of life. However, this approach may not work for phylogenetically distant strains. To extend the predictions to phylogenetically distant strains and to uncover genes that are mechanistically relevant for the trait of interest, machine learning models trained on gene content can be successful on some carbon sources. Leveraging biochemical knowledge can also enhance predictions by restricting the size of the feature space that the machine learning algorithm must consider. Prediction results from different models on the three datasets evaluated in this study are summarized in S1 Table.

Our findings emphasize the importance of testing statistical predictions out-of-clade. It is routine in machine learning to randomly hold out samples for testing. In situations like predicting traits of phylogenetically related organisms, it is necessary to include phylogenetic information when selecting strains to hold out. Our results emphasize that going forward, statistical methods should be evaluated using metrics that account for the phylogenetic structure of the data, such as prediction accuracy under out-of-clade data partition.

In this context, our study demonstrates that the most effective approach to predict phenotypes from genomes is through training statistical models on large libraries of isolates where quantitative phenotyping and genomic data are available. The example of tryptophan metabolism demonstrated the potential of such large datasets to potentially discover novel metabolic mechanisms. Results from synthetic data have demonstrated that datasets containing thousands of diverse strains can produce mechanistically insightful predictions. It is our hope that the results of this study help to spur the collection of large, quantitative, datasets of microbial traits. Emerging high-throughput phenotyping methods, such as robotics [26], droplet microfluidics [48], and culturomics [49], present exciting opportunities to acquire such data. Our study shows that datasets acquired with these technologies could enable highly accurate and interpretable predictions for metabolic traits of diverse bacterial taxa.

As demonstrated by the denitrification example above, statistical models may not yield detailed mechanistic insights into the pathway responsible for the phenotype of interest. Nevertheless, even if key predictors in models do not reveal underlying mechanisms, they are likely to exhibit strong co-occurrence patterns with genes necessary to perform target metabolic processes [45]. In these cases, statistical models can provide targeted hypotheses for further genetic or experimental investigations into the pathways underlying a given trait. However, if the objective is simply to accurately predict traits from genomes, more detailed mechanistic investigations are not necessary.

Our study predicts binary growth/no-growth traits from genomes. An important avenue for future work is to extend these approaches to predicting quantitative traits such as growth rates and biomass yields directly from genomes. It remains to be seen whether the approaches here will enable such predictions, but previous studies on mechanistically simple pathways [19] have demonstrated that such predictions are feasible. Larger datasets will likely be necessary for such predictions to succeed.

Extending the methods developed here to these quantitative traits would help us connect bacterial phenotypes to communities level processes. How communities function as a whole depends not only on the traits of individual strains but also on interactions between strains [19, 50, 51] and community assembly rules [25, 52–54]. We view the results here as an important first step to predicting emergent metabolic process rates in complex communities from genomic information alone. Accomplishing this goal would have broad impacts on predicting

metabolic process rates in important contexts ranging from the human microbiome [55] to soils [56].

## Methods

### Growth assay

We initiated the cultivation of each strain from frozen stocks in 0.75mL of 1/5X TSB medium (S2 Table) using a 48-well plate covered with a breath easier seal (USA Scientific Inc., #9126–2100) at 30˚*C* for a 48-hour period until saturation. Each strain was inoculated in two wells as two technical replicates. The cultures were then washed by centrifuging at 5200rpm for 10 minutes. We removed the supernatant and re-suspended the pellet with 1mL of carbon-free media (S3 Table).

The washed cultures were then grown in deep 96-well plates (Axygen PDW20C), where each column has media with a different carbon source and each row was inoculated with a different strain (Fig 1A). The growth medium consisted of the carbon-free medium (S3 Table) and 0.1M (carbon atom concentration) of a single carbon source, in a total volume of 1.2mL. The 12 columns consisted of 10 different media each with a distinct sole carbon source and 2 negative controls with carbon-free medium. Note that we added 0.05g/L yeast extract to the carbon-free medium to provide micronutrients to alleviate auxotrophies limiting growth, but at this concentration, yeast extract does not produce significant growth as measured by OD at 600nm. In each row, 10uL of a washed culture was inoculated into each well. Each plate includes 6 different strains along with 2 negative control rows without inoculated culture. The deep 96-well plate was covered with a sterile plastic cover (Thermo Scientific Nunc Edge 96-Well) and the edges were sealed using Parafilm (Bemis, PM996). The plates were incubated in a temperature-controlled room at 30˚*C*, shaken at 225rpm with 19mm orbit, for 72 hours.

After incubation, 150uL of culture from each well was transferred to a non-sterile flat-bottom 96-well plate (Greiner Bio-One 655101) and the optical density (OD) was measured at 600nm using a BMG Clariostar plate reader. The growth data was binarized by thresholding the final OD at 0.2, chosen based on the distribution of OD values of control and non-control groups (S1 Fig). Entries with inconsistent OD between the two technical replicates were removed.

### Sequencing and data processing

Out of the 96 strains, 86 strains were sequenced in previous studies [19, 24]. We obtained the assemblies of 65 strains directly from the author of [19] and 21 strains from NCBI [24].

We performed whole-genome shotgun sequencing on the additional 10 strains from [25]. The strains were grown in R2B media from frozen stock for 48 hours. We extracted DNA (Qiagen DNeasy UltraClean Microbial Kit, # 1224–250) and quantified the DNA concentration (Invitrogen Qubit dsDNA BR Assay Kit,# Q32853). For shotgun sequencing, we prepared the DNA library using the Illumina DNA Prep kit (# 20018704) and Nextera DNA CD Idx (96 Idx, 96 SPl) (# 20018708). The library was sequenced using the Illumina NextSeq 500/550 Mid Output KT v2.5 (300 CYS) (# 20024905) and demultiplexed using the NextSeq control software (version 4.0.2). The reads were trimmed using Trimmomatic (version 0.39, [57]) with parameters `ILLUMINACLIP:[ADAPTERFILE]:2:30:10:2:keepBothReads LEADING:3 TRAILING:3 MINLEN:36` (see supplementary file 1 for the custom adapter file, slightly modified from the built-in adapter file `NexteraPE-PE.fa`). Contigs were assembled using Unicycler (version 0.5.0, [58]), which utilized SPAdes (version 3.15.5, [59]).

All 96 strains underwent the same downstream analysis for functional annotation and 16S tree construction. We identified protein sequences from assemblies using Prodigal (version

2.6.3, [60]) and annotated KEGG ortholog groups using KofamScan (version 1.3.0, with KEGG release 104.0 [61]). We extracted 16S rRNA sequences from the assembly using Barnnap (version 0.9), constructed a multi-sequence alignment using SINA (version 1.7.2 [62], with SILVA release 138.1 SSU [63] as reference), and built a phylogenetic tree using FastTree (version 2.1.11 [64]).

The 96 strains contain complete KO, growth, and 16S data. The entire downstream analysis was organized as a Snakemake [65] pipeline, which is easily reproducible and available at https://github.com/zeqianli/CarbonUtilization.

## Constraint-based modeling

Using CarveMe (version 1.5.1, [17]), we constructed metabolic models from the protein sequence files generated by Prodigal (version 2.6.3, [60]). We built both non-gap-filled and gap-filled models. The built-in M2 minimum medium + glucose was used as the gap-filling media, as the recipe resembled the carbon-free media used in experiments (S3 Table).

We used a custom script to simulate growth on media with various carbon sources. In brief, given a metabolic model, we initialized the medium with the built-in carbon-free M2 minimal medium (the same recipe used for gap-filling) plus one of the ten carbon sources, enabling the uptake reaction of the carbon source, and we simulated growth in COBRApy (version 0.26.0, [66, 67]). To account for the potential misannotation of transporters, we included an option to force the carbon uptake reactions from the medium to the cytoplasm. Growth was binarized by thresholding the predicted biomass flux at 0.05 (mmol per gram dry weight of cells and hour). The choice of the growth threshold did not affect results (S2(D) Fig).

Results showed that gap-filling did not improve prediction accuracy, yet shifted the model behavior from overly specific (high false negative rate) to overly sensitive (high false positive rate) (S2(C) Fig). To test the effects of the gap-filling carbon source, we compared metabolic models gap-filled with different carbon sources. Our results showed that the gap-filling carbon source did not significantly affect the CBM prediction results (S2(B) Fig). Results also showed that forcing uptake did not affect prediction accuracies (S2(A) Fig).

## External datasets

The dataset from Gralka *et al.* was curated from [26], which provided whole-genome assemblies and growth rate measurements of 178 strains over 118 carbon sources. We annotated the KEGG orthology groups (KO) and built a phylogenetic tree using the same Snakemake [65] pipeline in this study. We binarized the growth data by assigning 1 to strains with non-zero growth rates and 0 to others. We then selected strains with complete KO, 16S rRNA, and growth data, and removed carbon sources with less than 10 entries of 1s and 0s in the growth data to ensure sufficient variance for prediction model evaluation. The final data consists of 172 strains and 100 carbon sources.

The BacDive dataset compiles utilization data from BacDive [39], annotated genomes from proGenomes v3 [68], and 16S rRNA sequences from the ENA database [69]. Raw data of strains with both utilization data and NCBI taxon IDs were downloaded from the BacDive download section. The data was parsed and utilization activities of "energy source" and "carbon source" were selected. To increase the sample size per metabolite, multiple forms for the same metabolite were merged (for example, "Arabinose", "L-Arabinose", and "D-Arabinose" were renamed "arabinose"). Although different enantiomers of metabolites may have different metabolic processes which could result in different utilization data, we found that such cases were rare (less than 0.5% of over 50, 000 records) and data entries resulting in conflicts in this merging process were subsequently removed. Therefore, phenotypes in the BacDive dataset

can be interpreted as the ability to utilize at least one form of the metabolite. Uncertain records (labeled as "+/-") and conflicting records were also removed.

To obtain genome data, we first linked the NCBI taxon ID in each BacDive entry with the NCBI `datasets` command line tools (version 14.5.1). Then, we match both IDs with the proGenomes v3 database [68] and download the EggNOG [70] annotation through its web API. To maintain annotation consistency with the other two datasets, we extracted the KEGG orthology group (KO) annotation from the EggNOG annotation files [29, 36, 70]. For coding sequences with multiple KO annotations, we selected the first one. Finally, we created a KO presence-absence matrix.

To obtain the 16S sequences, we retrieved rRNA sequences from the European Nucleotide Archive (ENA) FTP site [69]. We then linked the accession numbers with BacDive records to assign the 16S sequences. In cases where multiple 16S sequences were present in a BacDive record, we selected the longest one to ensure the completeness of the data. We generated a multi-sequence alignment of the 16S rRNA sequences using SINA (version 1.7.2 [62], with SILVA release 138.1 SSU [63] as the reference) and constructed a phylogenetic tree using FastTree (version 2.1.11, [64]).

Finally, we selected strains with complete KO, utilization, and 16S data. We selected carbon sources with utilization data from at least 100 samples and a balanced distribution of both 1s and 0s (at least 10 entries each) to ensure effective model evaluation. The resulting data contain 28553 utilization data entries, spanning across 4393 strains and 58 carbon sources. Note that sample sizes for different carbon sources are uneven, ranging from 104 to 2394 samples.

## Evaluating prediction models

For each model, we partition the dataset either randomly or out-of-clade (described in the next section) into a training set and a test set. We train models on the training set, predict on the test set, and calculate the prediction accuracy on the test set (ratio of correct predictions). We repeat this process for 100 partitions of the data and plot the distribution of accuracy scores on the test sets.

It is important to note that the prediction accuracy score, defined as the ratio of true predictions over the total number of predictions, is biased by the frequency of growth (1) and no-growth (0) for that carbon source (trait). For traits with a large fraction of either zeros or ones, over-specific and over-sensitive models, which predict mostly zeros or ones, can yield high prediction accuracy despite poor predictions. To account for this bias, when evaluating model accuracy, we always compared the distribution of prediction accuracy across the test sets to two null models:

1. Bernoulli null model: from the training set, the Bernoulli probability $p$ is calculated by the fraction of 1s in the training set (for each partition of the data). For each test strain, the null model predicts 1 with probability $p$ and 0 by probability $1 - p$. This null model shows a baseline of prediction accuracies of guessing by chance.

2. Identity null model: find the more common phenotype in the training set (1 or 0). When predicting for test trains, always predict this phenotype. This null model shows a baseline of prediction accuracy by models that are overly specific or overly sensitive.

When comparing two models, we compare model accuracy scores by using a permutation t-test (two-sided, $10^5$ permutations; implemented using the `ttest_ind` function in the `scipy` package (version 1.9.3, [71]). For CBM, we computed the p-value by comparing the CBM prediction accuracy against the background of null prediction accuracies: we compute a distribution of prediction accuracies generated from null models (Bernoulli or identity) over

$10^5$ random data partitions and the p-value is calculated as the ratio the null model predictions with accuracy scores higher than the CBM prediction accuracy over all null model predictions. The p-values are adjusted to account for multiple testing on the 10 carbon sources using the Holm-Sidak correction [72, 73], implemented by the `multipletests` function in the `statsmodels` package (version 0.13.5).

It is important to note that a model that does not significantly outperform both null models does not necessarily indicate its failure. This could simply be due to a lack of variability in the distribution of zeros and ones in the test sets, causing the identity null model to be highly accurate. In such cases, the dataset lacks sufficient variance in the trait and the data is not sufficient to accurately assess the model's performance. To better differentiate a prediction model from the null models, additional data with a more balanced fraction of ones and zeros is necessary.

### Partitioning data into test and training sets

When evaluating a statistical model, we partition a complete dataset into a training set and a test set. This partition is performed via two methods:

- Random partition: randomly select 20% of the samples as the test set and the remaining 80% as the training set.

- Out-of-clade partition: The aim here is to choose a test set that is phylogenetically distinct from the training set while maintaining comparable test set sizes, (10% to 30% of the data). The steps are following:

  1. Iterate all clades (sub-trees) in the phylogenetic tree. Identify clades containing node numbers within the target size range (10% to 30% of all samples). These clades are options for the test set.

  2. To create more test set options, we allow test sets to comprise of more than one clade. We merge random pairs of clades in step 1 and retain pairs with the total node numbers within the target test set size range (10% to 30% of all samples). These clade pairs as additional options for the test set.

  3. For each out-of-clade partition, randomly select a test set option. The remaining samples form the training set.

### Nearest-neighbor

In the nearest-neighbor model, a test sample's phenotype is predicted by the same phenotype of its phylogenetically closest strain in the training set or the strain with the most similar genome, determined by a distance metric. The model was implemented using `KNeighborsClassifier` from the `scikit-learn` package (version 1.1.3, [74]), with the parameter `n_neighbors = 1`. When using 16S sequences as predictors, we used the Hamming distance (metric = 'hamming') (S5(A), S5(B) and S10(A) Figs), except for the BacDive dataset, where we used branch distances on the phylogenetic tree when predicting by phylogeny, due to computational limitation (S11(B) Fig). When using KO vectors as predictors, we used the $L_1$ distance between vectors of KO presence/absence (Figs 2B, 3B and S11(C)).

### Random forest

The random forest model was implemented using `RandomForestClassifier` from the `scikit-learn` package (version 1.1.3, [74]). Results in the main figures used default

parameters except `max-features = None`, which allow each decision tree to utilize all features instead of a subset of features. This modification was necessary as we discovered that subsampling features resulted in suboptimal performance because key predictors could be omitted in some decision trees.

To study the effects of hyperparameters, we tuned `max_depth`, which limited the number of features used for the prediction by limiting the decision tree depth, and `max_features`, which determined whether each decision tree uses all features (`max_features = None`) or a subset of features (`max_features = "sqrt"`). We found tuning hyperparameters do not improve model prediction accuracies (S4 Fig).

## Phylogenetic correlation length scale

Phylogenetically close strains are more likely to exhibit the same trait up to a certain evolutionary distance (S5(C) Fig). We utilized chi-square statistics to quantify the phylogeny-trait correlation length scale for each carbon. Intuitively, we wanted to identify a phylogenetic distance, beyond which strains are no longer more likely to have the same traits compared to a null model of randomly distributed traits. The steps are as follows:

1. Calculate the phylogenetic tree distances between all pairs of strains, defined by the total branch lengths between two nodes on the phylogenetic tree generated by FastTree [64].

2. Iterate over a range of tree distance values ($d$) and collect all strain pairs with phylogenetic distances within the sliding window [$d - 0.1, d$). Changing the interval size (0.1) did not impact the results.

3. Count the number of strain pairs within and outside the interval with the same or different traits (growth/no-growth on the carbon source). Construct the following contingency table:

|  | Same trait | Different trait |
| --- | --- | --- |
| Within the window |  |  |
| Outside the window |  |  |

4. Compute the $\chi^2$ statistics (implemented using the `chi2_contingency` function in the `scipy` package (version 1.9.3, [71])).

5. The largest $d$ value with significant chi-square statistics ($p < 0.05$) is defined as the phylogeny-trait correlation length scale.

If the correlation is not detected for a carbon source, i.e., the $\chi^2$ statistics for the smallest $d$ value is not significant, we assign the value 0 as its correlation length scale.

We calculated the phylogeny-trait correlation length scale for the 10 carbon sources in this study and compared it with the phylogenetic distances between test strains and their training nearest-neighbor strains, for both random and out-of-clade cross-validation (S5(D) Fig). We find that when the test set is held out randomly, for most test strains, there is a phylogenetically nearby neighbor, whose phylogenetic distance is much less than the correlation distance for each trait. As a result, predicting the same trait as the nearest neighbor is by and large accurate. Yet under the out-of-clade data partition regime, we found that test samples and their training nearest neighbors are often more distant than the correlation length, accounting for the failure of nearest-neighbor classifiers in this case (S5(D) Fig). Similar results were observed for based on the dataset curated from [26] (S10(B) and S10(C) Fig). Yet for the dataset curated from

BacDive [39], most strains are phylogenetically distant (S11(D) Fig) and even for randomly partitioned training and test sets, nearest strains are phylogenetically further apart than the phylogeny-trait correlation length (S11(D) and S11(E) Fig). This explains the poor prediction performance based on 16S sequences for most carbon sources (S11(B) Fig).

This result solidified the idea that predictions relying purely on phylogenetic relatedness require sufficient proximity between the target strain and the reference strain.

## Understanding random forest's behavior under random and out-of-clade dataset partitions

Random forest classifiers trained using gene content as predictors can accurately predict carbon utilization when the dataset is partitioned randomly but it often fails for out-of-clade partitions of test and training sets. We hypothesize that the random forest models trained in random partitions make predictions by exploiting the phylogeny-trait correlation, similar to nearest-neighbor models.

During cross-validation, many training/test partitions were generated (random or out-of-clade). We use $i$ to denote each dataset partition, with training samples $\mathbf{X}^i = \{\mathbf{x}_s^i\}$, $Y^i = \{y_s^i\}$ and test samples $\tilde{\mathbf{X}}^i = \{\tilde{\mathbf{x}}_s^i\}$, $\tilde{Y}^i = \{\tilde{y}_s^i\}$, where $\mathbf{x}$ is a binary vector indicating gene presence-absence and $y \in \{0, 1\}$ indicates traits. The bold font for $\mathbf{x}$ indicates vectors and the regular font for $y$ indicates binary values.

A model is trained using the training data, resulting in a mapping function $f^i : \mathbf{x}_s \rightarrow f^i(\mathbf{x}_s) \in \{0, 1\}$. We ask whether $f$ resembles the nearest neighbor classifier based on $L_1$ distance of gene presence-absence vectors. For each $\tilde{\mathbf{x}}_s^i \in \tilde{\mathbf{X}}_s^i$, we find its $L_1$ nearest neighbor in the training set:

$$\mathbf{x}_s^{nn,i} = \arg \min_{\mathbf{x} \in \mathbf{X}^i} L_1(\tilde{\mathbf{x}}_s^i, \mathbf{x})$$

Then, we compute the difference of the predicted values for $\tilde{\mathbf{x}}_s^i$, $\mathbf{x}_s^{nn,i}$ averaged across all test samples in data partition $i$:

$$\mu^i = \mathbb{E}_{\tilde{\mathbf{x}}_s^i \in \tilde{\mathbf{X}}^i} \left( |f^i(\tilde{\mathbf{x}}_s^i) - f^i(\mathbf{x}_s^{nn,i})| \right)$$

If $f$ were a nearest-neighbor classifier using $L_1$ distance, $\mu^i = 0$. For other models, if the distribution of $\mu^i$ across dataset partitions ($i$) is close to 0, we consider it to behave similarly to the nearest neighbor classifier based on $L_1$ distance for gene presence-absence. To do this, we compare the distribution of $\mu^i$ to a null model where we calculate a similar value but using randomly chosen training samples instead of the nearest neighbors:

$$\mathbf{x}_s^{rand,i} : \text{randomly sampled from } \{\mathbf{x}_s^i\}$$

$$\mu^{rand,i} = \mathbb{E}_{\tilde{\mathbf{x}}_s^i \in \tilde{\mathbf{X}}^i} \left( |f^i(\tilde{\mathbf{x}}_s^i) - f^i(\mathbf{x}_s^{rand,i})| \right)$$

S6(D) Fig shows the distribution of $\mu^i$ and $\mu^{rand,i}$ across 100 data partitions (random or out-of-clade) for the 10 carbon sources in this study. Results show that the trained random forest models, on both random partitions (left) and out-of-clade partitions (right), are very likely to make the same predictions for a test sample and its $L_1$ training nearest neighbors. Consequently, random forests behave similarly to nearest-neighbor models using the $L_1$ gene vector distance. And we showed that the nearest-neighbor classifiers using $L_1$ gene distance, similar to using the hamming distance of 16S sequences, can make accurate predictions when the dataset is partitioned randomly but not out-of-clade (Fig 2B). This is because, for strains in

this study, genome similarity ($L_1$ distance of gene presence-absence vectors) strongly correlates with 16S similarity (S6(A) Fig), agreeing with other observations in the field [20, 75].

In fact, this effect extends to other statistical models with a mapping function $f$ that is likely to map similar inputs to the same response value. The random forest classifier is an example because samples with similar genomes are likely to go through similar decision paths and consequently receive the same predicted outcome. To test this hypothesis, we repeat the analysis above on another example of such classifiers, logistic regression. Logistic regression is implemented using `LogisticRegression` in the `scikit-learn` package [74], with $L_1$ regularization using parameters `C = 1.0, penalty = 'l1', solver = 'liblinear'`. Similar to the random forest, on both random partitions and out-of-clade partitions, the trained logistic regression classifiers are likely to map test samples and their $L_1$ training nearest-neighbors to the same predicted outcome compared to the null model, consequently behaving like nearest-neighbor models (S6(E) Fig). We saw that indeed, the logistic regression classifiers can predict traits accurately for random partitions and fail for the out-of-clade partitions (S6(B) and S6(C) Fig).

This observation shows that making accurate predictions on the random partition is easy due to the phylogeny-trait correlation: as long as the mapping function generates similar predictions for similar genomes, the functional form of a model does not matter. To break this pattern of accurate predictions on random partitions but not out-of-clade, we hypothesize that changing the functional form of the model is insufficient. Additional steps, such as feature selection and increasing sample size, are necessary.

## Feature selection by conditional entropy

To overcome the large number of features in the prediction of traits from gene content (KOs), we attempted feature selection to restrict predictors to only a few that are most predictive of the trait in the training set. We defined the predictive power of a set of features (KOs) using conditional entropy.

Consider that a small set of $k$ genes (out of $n$ genes in total) is selected as predictors, which comprised the predictor matrix $X$ of shape $(m, k)$, where $m$ is the total number of samples and $X_{ij}$ indicates the presence/absence of $j$th gene in $i$th sample. Denote the trait vector as a binary vector $y$ of length $m$, where $y_i$ indicates growth on $i$ sample.

Let $\{x_s\}$ indicate all unique gene vectors (rows) in $X$. The conditional entropy is given by

$$H(y|X) = \sum_s P(x = x_s)H(y|x = x_s) \tag{1}$$

$$= -\sum_s P(x = x_s) \sum_{t \in \{0,1\}} P(y = t|x = x_s) \log P(y = t|x = x_s) \tag{2}$$

Large conditional entropy indicates large uncertainty in growth ($y$) given the same feature vector (rows in $X$), while $H(y|X) = 0$ when there is a one-to-one map from feature vectors in $X$ to $y$ and a perfect prediction is theoretically possible. Therefore, conditional entropy measures the upper limit of a model's predictive power.

Given the full KO matrix in the training set, we find the submatrix consists of maximal $k$ columns in $X$ that minimize the conditional entropy. We chose $k \leq 5$ because in our dataset including more genes does not reduce conditional entropy further (S8 Fig). Enumerating all combinations of $k$ genes is computationally infeasible, so we used a greedy search procedure:

1. Given a training set and a test set partition by the out-of-clade partition rules, the objective is to select $k = 5$ genes with the most predictive power in the training data.

 

2. In the training data, calculate the conditional entropy of all single genes. Pick the top 5 genes with the smallest conditional entropy. We kept the top 5 contenders for the next step to reduce the greediness in the search process.

3. Pair one of the 5 genes from the previous step with a different gene. Calculate conditional entropy for all pairs of genes and pick the top 5 gene pairs with the smallest conditional entropy.

4. Repeat 2 and keep adding genes until the number of selected predictors reaches 5.

This process is repeated over 100 out-of-clade partitions. We observed that feature selection did not improve random forest prediction in out-of-clade partitions (S7 Fig). Changing the number of selected features did not improve prediction results (S9 Fig).

To investigate the feature selection process, we ask what the distribution of conditional entropy on possible gene combinations is. Specifically, on the full dataset, for each carbon source and each $k \in \{1, 2, 3, 4, 5\}$, we pick the top 100 combinations of $k$ genes with the lowest conditional entropy and plot the distribution (S8 Fig). For most carbon sources, when $k \leq 2$, there is one distinct choice of features that encode most information of the trait, yet the encoded information is not perfect (as the entropy is larger than 0). As $k$ increases, all most all 100 choices of the $k$ genes are equally predictive of the trait. As a result, feature selection did not improve the random forest's out-of-clade prediction accuracy, regardless of the number of selected features ($k$) (S9 Fig): when the $k$ is small, the selected features are not sufficient to encode the trait; yet when $k$ is large, the choice of the best features in the training data is highly redundant, yet not all of them are predictive on the test data. Because a model cannot distinguish these choices using just the training data, choosing an arbitrary one often led to poor prediction on the test data.

## Meta-learning

Meta-learning is a method in the field of domain generalization to design models predictive to a new domain of data [34]. The training set is divided into meta-training and meta-test sets as different data domains, where models are tuned until the resulting models have cross-domain predictive power. The problem of making predictions out-of-clade is identical to domain generalization, in the sense that we want the algorithm to learn features that generalize to new clades (domains). Therefore, we adopted this procedure to identify features that can generalize out-of-clade using a similar greedy search process:

1. Given a training set and a test set partitioned by the out-of-clade partition rules, the objective is to select $k = 5$ features with cross-domain (out-of-clade) predictive power using only training data.

2. Divide the training set into meta-training and meta-test sets using the out-of-clade partitioning rules. Generate 20 partitions.

3. For each gene (columns in the KO matrix), train a binary classifier using the gene as the only predictor on the meta-training sets and test on the meta-test sets. Pick the top 5 genes with the highest mean accuracy on meta-test sets across meta-partitions.

4. Pair one of the 5 genes with a different gene. For each pair, train a decision tree using them as features on meta-training sets and test on meta-test sets. Pick the top 5 gene pairs with the highest mean accuracy on meta-test sets across meta-partitions.

5. Repeat until 5 genes are selected as predictors. Pick the 5-gene combination with the highest mean accuracy across meta-partitions.

6. Finally, train a decision tree using the final 5 genes on the full training set. Evaluate the model prediction accuracy on the true test set.

### Generating synthetic traits to computationally determine the role of sample size and gene number on predictions

In Fig 5E we generated synthetic traits using real genomes from the BacDive dataset. Using real genomes allowed us to capture realistic gene co-occurrence patterns, gene frequencies, and phylogenetic correlations. A synthetic trait is determined by the presence-absence of a set of genes, as illustrated in Fig 5D. To study how properties of the dataset affect a model's behavior, we generated synthetic datasets with three different parameters: the number of samples (100, 250, 500, 1000, 4349), tuned by randomly sub-sampling the genomes; the number of predictors (100, 300, 500, 2000, 8479), tuned by randomly sub-sampling the KOs; and the number of trait-defining genes (3, 10, 30, 100, 300), tuned by using different numbers of genes when generating the synthetic trait. Additionally, to capture the modular nature of real biochemical traits, we introduced a modularity parameter in defining synthetic traits. Details are as follows:

1. Given the number of genomes and the number of predictors, sub-sample the strain collection and KOs from the KO matrix in the BacDive data.

2. Given the number of trait-defining genes, randomly select them from the remaining KOs.

3. Define the synthetic trait as a series of random logic operations AND/OR among the trait-defining genes. To capture the modular structure of metabolic pathways (e.g. a metabolic process often consists of multiple steps, with each step enabled by a set of enzymes), we introduce a modularity parameter in the generation process. Intuitively, $(g_1\ AND\ g_2)\ OR\ (g_3\ AND\ g_4)$ is more modular than $((g_1\ AND\ g2_2)\ OR\ g_3)\ AND\ g_4)$. In the implementation (see the source code for details), the logic operation is modeled as a randomly populated binary tree. The modularity parameter controls the minimum ratio of genes in the left and right sub-trees in every node in the generation process. We generated synthetic traits with modularity 0 and 0.3.

4. Generate 50 synthetic traits for the genomes based on the presence-absence of genes, with each trait including at least 10% of 0s and 1s for effective model evaluation.

We partitioned each synthetic dataset into a training set and a test set by either the random or the out-of-clade partition rules, trained models on the training data, and tested on the test data. We repeated the process on 100 data partitions and all 50 synthetic traits. Figs 5E, S12 and S13 plot the distribution of prediction accuracy under different dataset properties, with the line indicating the mean accuracy score across all partitions and traits and the shade indicating the 95% confidence interval.

Results demonstrated that random forests achieved significantly higher prediction accuracy on randomly partitioned data compared to out-of-clade partitions, as observed in real-world data (compare Figs 5E and S1(A)), both significantly outperformed null models (S12(B) Fig). Notably, the accuracy of random forests on out-of-clade partitions generally increased with larger sample size (y-axis in Fig 5E), fewer predictors (compare panels in Fig 5E), and less complex traits (compare colors in Fig 5E). The trend was consistent when evaluating predictions using balanced accuracy, which is defined as the arithmetic mean of sensitivity and specificity (S12(C) Fig), indicating that accurate prediction is not an effect of unbalanced trait data. These findings held true when the synthetic traits were designed to be modular and overall

prediction accuracies were slightly lower on modular traits compared to non-modular ones (compare Figs 5E, S12 and S13).

## KEGG data

We utilized the KEGG pathway database [29, 35, 36] to select genes associated with metabolites through an automated process. We used the KEGG API to download KEGG reactions, KO, and compound data in pathways related to central carbon metabolism, carbohydrate metabolism, and lipid metabolism. For each carbon source, we identified relevant KOs with three different levels of specificity (Fig 4A):

1. Pathways: All KOs in pathways that included the carbon source.

2. Paths to the central metabolism: our objective was to, through an automated process, identify a set of genes that metabolize the supplied carbon source to the central carbon metabolism. We processed KEGG reaction data and created a reaction graph where the nodes represent metabolites and the edges represent reactions. Given a carbon, the top 3 shortest paths connecting it to a list of destination compounds (glucose, fructose, glycolysis intermediate compounds, pyruvate, TCA intermediate compounds; see supplementary data for the full list) are identified. The KOs involved in these reactions were selected. It is worth noting that the KEGG reactions do not always specify the reaction direction. Hence, we repeated the process with both a directed reaction graph (where each reaction is directional from the first compound to the second compound in the KEGG reaction entry) and an undirected graph (where each reaction is considered bidirectional). Both graphs yielded similar results and we used the directed graph for results in Fig 4B.

3. Reactions: All KOs in reactions that included the carbon source.

The list of KOs selected through the three methods is included in the supplementary data.

## Predicting nitrite reduction for denitrifiers

A previous study characterized 54 of the 96 strains in our dataset in the context of denitrification [19]. Denitrification is a four-step process whereby nitrate is reduced sequentially to nitrite, nitric oxide, nitrous oxide, and eventually nitrogen gas, with each step being catalyzed by a set of well-characterized enzymes (S14(A) Fig). A microbial strain can perform either one or multiple steps.

Under anaerobic conditions, 53 out of the 54 strains can reduce nitrate and 37 can reduce nitrite (S14(A) Fig). We asked if statistical methods can predict whether a strain could reduce nitrite or not. To test this, we trained random forest classifiers using the full genome (KO presence-absence) as predictors to predict nitrite reduction. We evaluated model prediction accuracies under both random and out-of-clade partitions (S14(B) and S14(C) Fig).

We found that random forests predicted the trait near perfectly for both partitions (S14(B) and S14(C) Fig). Interestingly, the success of the random forest models was mainly attributed to the nitric oxide reductase, *qNor*, which had the highest feature importance (S14(D) Fig)), instead of the nitrite reductases, *nirS* and *nirK*, which are the enzymes that mechanistically enable nitrite reduction (the trait being predicted). This is because the *qNor* gene vector is almost identical to the vector *nirS* OR *nirK*, both of which almost perfectly encoded the nitrite reduction trait. This is a known observation that, because nitric oxide is toxic, nitrite-reducing bacteria (which produce nitric oxide) nearly always reduce nitric oxide as well [44]. Without this knowledge, statistical models opted for using fewer features (*qNor*) when making predictions.

## Supporting information

**S1 Fig. Choosing OD threshold to determine growth/no growth.** Distribution of OD (600nm) measurements after 72-hours of incubation. The left panel shows the OD values of control groups, including wells with either no carbon source in the medium or no inoculated culture. The right panel shows OD of experimental groups, where wash cell cultures are inoculated to media with a carbon source. The red dashed line indicates the growth/no-growth threshold (0.2), where we observed a clear cutoff of the control group wells.
(TIF)

**S2 Fig. Gap-filling and forced carbon source uptake do not improve constraint-based model prediction accuracy.** (A) Gap-filling or forcing uptake did not improve CBM prediction accuracy. Grey violin plots: distribution of accuracy of the Bernoulli null model. CBM predictions with significantly higher accuracy than the null model after multiple-testing correction (Methods) are marked *. (B) The choice of carbon source used for gap-filling did not significantly alter the prediction outcome. The left panel shows the prediction accuracy of CBM models gap-filled with one carbon source (x-axis) on the growth with another carbon source (y-axis). The right panel shows the same results while forcing uptake reactions for the predicted carbon source from the medium to the cytoplasm. Entries with the same gap-filling carbon and prediction carbon are omitted because these metabolic models always predict positive growth. (C) Sensitivity and specificity of CBM models. While gap-filling or not yielded similar prediction accuracy, non-gap-filled CBM models had a high false negative rate (low sensitivity) while gap-filled CBM models had a high false positive rate (low specificity). (D) Distribution of CBM predicted growth, defined as the flux through the biomass reaction. Red line: our choice of growth threshold (0.05 mmol per gram dry weight of cells and hour). Changing the growth threshold did not significantly affect the prediction results.
(TIF)

**S3 Fig. Feature importance of random forest models trained under randomly partitioned training and test sets.** Gene feature importance scores for the 10 carbon sources in random forests trained under random data partitions (blue in Fig 2B). For each gene, the feature importance score in the y-axis was calculated as the mean feature importance score of 100 random forests trained in the 100 random data partitions. Gene functions are highlighted by color and were assigned using KEGG pathways (see Methods). Note that although random forests show high prediction accuracy in Fig 2B, these predictions often did not rely on enzymes related to the metabolism of the target carbon source.
(TIF)

**S4 Fig. Varying hyperparameters in random forest classifiers did not improve prediction accuracy.** We tuned two key hyperparameters in random forests models: maximum tree depth and feature number used per decision tree (Methods). Varying these hyperparameters did not improve random forest predictions in both randomly held out test sets (top) and out-of-clade test sets (bottom).
(TIF)

**S5 Fig. Predicting carbon utilization traits from phylogenetic information alone.** (A) Trait prediction accuracies by nearest neighbor classifiers using 16S rRNA distances (hamming distance; see Methods) as predictors (blue) with randomly partitioned training/test sets as in Fig 2A. Predictions are tested against two null models (Bernoulli, identity; see Methods), and those that significantly outperform both null models are marked * ($p < 0.05$ after a multiple-testing correction; see Methods). (B) Trait prediction accuracy for out-of-clade test sets (as in

Fig 2C), using nearest neighbor classifications on 16S rRNA sequence (green). Predictions that significantly outperform both null models are marked * ($p < 0.05$). (C) Phylogenetic correlation of the arabinose utilization trait. X-axis is the phylogenetic distance between strains. Y-axis is the average difference in trait below the distance, calculated as follows: at a given phylogenetic distance, we identify all strain pairs with distances below the value and calculate their mean difference of traits (1 for different traits and 0 for identical trait—growth/no-growth). The blue curve shows the mean trait difference as a function of the phylogenetic distance in our data. The red curve shows the expected curve of a null model with randomly distributed traits and the red-shaded region indicates the standard deviation over 100 bootstrapping instances. The green dashed line indicates the correlation length scale for arabinose calculated using chi-square statistics (Methods). (D) The correlation length scale for the 10 carbon sources (green markers). Violin plots show the distribution of phylogenetic distances between test strains and their training nearest neighbors for random partitions of the data (blue) and out-of-clade partitions (orange).
(TIF)

**S6 Fig. Random forest classification using gene presence/absence predicts traits by exploiting phylogenetic correlations.** (A) The genome similarity, defined by the $L_1$ distance of gene presence-absence vectors for a pair of strains (x-axis), and the phylogenetic distance, defined by the hamming distance between 16S rRNA sequence for a pair of strains (y-axis), for all strain pairs. The genome distance strongly correlates with the phylogenetic distance. Therefore, nearest-neighbor models using gene presences-absence performed similarly to those using 16S rRNA sequences (compare Figs 2B, 2C and S5(A) and S5(B)). (B)(C) Logistic regression, like the random forest, predicts traits accurately under random data partitions (B) but not for out-of-clade partitions (C). (D) Comparing random forests' prediction on test samples and their training nearest neighbors. We compute the mean predicted trait difference between test samples and their $L_1$ training nearest-neighbors, on both random partition (green, left panel) and out-of-clade partition (blue, right panel, see Methods for details). Compared to a null model (grey) of picking a random strain in the training set instead of the nearest-neighbor by genome similarity ($L_1$ distance of gene presence-absence vectors), RF tends to make the same prediction for a test sample and the training sample with the most similar genome, consequently behaving like a nearest-neighbor model using $L_1$ genome distances. Bars on the top indicate p-values, computed by permutation t-tests (two-sided, $10^5$ permutations) and corrected for multiple testing on 10 carbon sources (Methods). (E) Similar to random forests in (D), logistic regression also tends to make the same predictions for test samples and their training nearest neighbors, for both random (light orange, left panel) and out-of-clade partitions (dark orange, right panel). see Methods for details.
(TIF)

**S7 Fig. Random forest classification using feature selection by conditional entropy or meta-learning.** The greedy feature selection method (pink; Methods) and the meta-learning approach (orange; Methods) were tested on out-of-clade test sets. Each model is cross-validated over 100 data partitions and plots show the distribution of prediction accuracy. Predictions that significantly outperform both null models at the same data partition are marked * at the top ($p < 0.05$ after multiple-testing corrections; see Methods.)
(TIF)

**S8 Fig. Conditional entropy as a function of selected gene numbers.** Violin plots show the distribution of conditional entropy (Methods) between the top 100 combinations of one to five genes with the lowest conditional entropy. The combination with the lowest conditional

entropy is marked as the red cross. Note that as the number of genes increases, the vertical extent of the violin plot decreases, indicating the top 100 combinations all have similar conditional entropy with the trait. This means that when 5 genes are selected, all 100 choices of the 5 genes are equally predictive of the trait. In contrast, for only one gene the best predictor (red "x") is unique in that it is an outlier in the distribution, but this gene presence/absence has high conditional entropy with the target trait and is therefore a poor overall predictor. We argue that this redundancy effect, as a result of the small number of samples and the large number of predictors, prevents effective feature selection.
(TIF)

**S9 Fig. Random forest classification using feature selection by conditional entropy with different numbers of selected genes.** Prediction accuracy of the random forest classification with different numbers of selected genes, under out-of-clade partitions with different numbers of features. Changing the number of selected features did not improve the model's performance.
(TIF)

**S10 Fig. Prediction accuracies for the dataset curated from Gralka *et al.* [26].** (A) Using 16S sequences alone as predictors, the phylogenetic nearest-neighbor classifier successfully predicted growth on 76 out of 100 carbon sources when the dataset is partitioned randomly yet only 24 out of 100 carbon sources when the dataset is partitioned out-of-clade. A prediction is considered successful when the accuracy over 100 data partitions is significantly higher than the two null models by a permutation t-test ($10^5$ permutations) after multiple testing corrections for the 100 carbon sources ($p < 0.05$; see Methods). (B)(C) The empirical cumulative distribution function (ECDF) of the phylogeny-trait correlation length scale calculated for the 100 carbon sources (see S5(D) Fig and Methods). If we fail to detect correlation, the length scale value is assigned 0. (C) list the phylogeny-trait correlation length scales for all 100 carbon sources. The 14 carbon sources where RF outperformed KO nearest-neighbor models in out-of-clade prediction (Fig 3C) are highlighted with *. Their correlation length scales are shorter than the other carbon sources (p = 0.03, permutation t-test). (D) Comparison of random forest out-of-clade prediction on isoleucine utilization using gene presence-absence, without (blue) or with feature selection using KEGG (orange). While both predictions are significant compared to the two null models (grey), the feature selection using KEGG by restricting predictors to only genes interacting with the carbon compound significantly improved the prediction accuracy (compare orange to blue; $p = 0.0036$, permutation t-test with $10^5$ permutations after multiple-testing correction.) (E) Feature importance scores for random forests trained in panel (D) without (left) or with (right) feature selection. Features selection forced the random forest to use key metabolic enzymes instead of using phylogenetic signals (note the gene with a high feature importance score is an enzyme that interacts with isoleucine), resulting in improved prediction accuracy.
(TIF)

**S11 Fig. Predicting carbon utilization traits for the BacDive dataset [39] using phylogenetic information alone.** (A) The curated dataset consists of binary utilization data spread across 4349 diverse microbes on 58 carbon sources, with sample size for each carbon source ranging from 104 to 2394 samples. The matrix shows growth/no-growth data as in Fig 1. (B) Using 16S sequence distances to predict traits. The phylogenetic nearest-neighbor model using only 16S sequences as predictors outperformed both null models (Bernoulli and identity) on only 4 out of the 58 carbon sources for the random partition and none for the out-of-clade partition. (C) Use KO presence-absence to predict traits. When the training sets and test sets are

partitioned randomly, random forest outperformed both null models (Bernoulli and identity) on 13 (blue and orange) out of the 58 carbon sources and 6 carbon sources (blue) were also significantly predicted by nearest neighbor classifiers. When the dataset is partitioned out-of-clade, only 2 carbon sources were significantly predicted by random forest (orange) and none was significantly predicted by nearest-neighbor. (D) Empirical cumulative distribution function (ECDF) of the phylogeny-trait correlation length scale for the 58 carbon sources. We failed to detect correlation for 45 out of the 58 carbon sources, for these carbon sources we assigned a value of 0. (E) Comparing train-test nearest-neighbor distances with the phylogeny-trait correlation length scale for the 13 carbon sources with measurable phylogeny-trait correlation. Similar to S5(D) Fig, for each carbon source, we divided the available data either randomly or out-of-clade and, for each sample in the test set, found its phylogenetically closest neighbor in the training set and calculated their phylogenetic distance (the branch distances between the two nodes on the tree). For each carbon source, we compared the distribution of the train-test nearest neighbor distance (y-axis; point shows the mean and the error bar shows the 50% percentile interval) with the phylogeny-trait correlation length scale (x-axis) detected for the carbon source. Unlike the dataset in this study, for most carbon sources, strains available in the BacDive dataset are phylogenetically distant and the trait-test nearest-neighbor distances are further than the correlation length, even in the random partition. The 4 carbon sources with significant prediction accuracy from the nearest-neighbor model under random partitions are highlighted in the legend.
(TIF)

**S12 Fig. Random forest prediction results on synthetic data with non-modular traits.** (A) Random forest prediction accuracy for synthetic traits under random partitions with synthetic datasets with different properties, including different sample sizes (x-axis), different numbers of predictors (different panels), and different trait complexities (different colors). Note the higher prediction accuracy compare to the out-of-clade prediction accuracy in Fig 5E. (B) Prediction accuracy of the Bernoulli null model and the identity null model under random and out-of-clade partitions. Note the lower prediction accuracy compare to panel (A) and Fig 5E. (C) Balanced accuracy, defined as the arithmetic mean of sensitivity and specificity, of random forest predictions on out-of-clade partitions (same scenario as Fig 5E). Note the high balanced accuracy score, which indicates that the high accuracy score in Fig 5E is not an effect of unbalanced fractions of zeros and ones in the trait data.
(TIF)

**S13 Fig. Random forest prediction results on synthetic data with modular traits.** The model prediction accuracy on synthetic data with traits at modularity 0.3 (see Methods and the supplementary code for details). Note that for synthetic datasets with the same sample size, the same number of predictors, and the same number of trait-defining genes, predicting modular traits (this figure) has a lower prediction accuracy than predicting non-modular traits (modularity = 0; Figs 5E and S12). Each panel shows similar information as Figs 5E and S12:
(A) Random forest prediction accuracy for synthetic datasets under out-of-clade partitions.
(B) Random forest prediction accuracy for various synthetic datasets under random partitions.
(C) Bernoulli null and identity null prediction accuracy on random or out-of-clade partitions.
(D) Balanced accuracy of random forest prediction on out-of-clade partitions.
(TIF)

**S14 Fig. Statistical prediction of the nitrite reduction trait for bacterial denitrifiers.** (A) Presence-absence of denitrification-related genes of 54 strains (left matrix) and their nitrite reduction capability (right vector) from Gowda *et al.* [19]. Blue: nitrate reductases; orange:

nitrite reductases; green: nitric oxide reductase; red: nitrous oxide reductase; pink: regulators; brown: transporter. Gene colors correspond to different reactions in the denitrification cascade illustrated at the bottom. Note that the gene vector (*nirS* or *nirK*) is almost identical to the *qNor* gene vector, which is almost identical to the nitrite reduction trait (NIR, last column). (B)(C) Prediction of whether strains grow on nitrite (via anaerobic respiration) from the full genomes using nearest-neighbor models and random forest models for both random (B) and out-of-clade partitions (C). (D) Mean feature importance of genes in the random forest models trained out-of-clade partitions. The nitric oxide reductase, *qNor*, is used as the strongest predictor in random forests.
(TIF)

**S1 Table. Summary of prediction results.**
(PDF)

**S2 Table. Modified 1/5x TSB medium composition.**
(PDF)

**S3 Table. C-free medium composition.**
(PDF)

## Acknowledgments

We acknowledge useful discussions with Arvind Murugan, Yuxin Chen, Matti Gralka, Mikhail Tikhonov, and Tong Wang. We acknowledge the Research Computing Center at the University of Chicago for providing computational resources.

## Author Contributions

**Conceptualization:** Zeqian Li, Ahmed Selim, Seppe Kuehn.

**Data curation:** Zeqian Li, Ahmed Selim.

**Formal analysis:** Zeqian Li, Ahmed Selim.

**Funding acquisition:** Seppe Kuehn.

**Investigation:** Zeqian Li.

**Methodology:** Zeqian Li.

**Project administration:** Seppe Kuehn.

**Resources:** Zeqian Li.

**Software:** Zeqian Li.

**Validation:** Zeqian Li.

**Visualization:** Zeqian Li.

**Writing – original draft:** Zeqian Li, Seppe Kuehn.

**Writing – review & editing:** Zeqian Li, Seppe Kuehn.

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
