## [Decision Letter · Decision Letter 0]

19 Sep 2023

Dear Dr. Kuehn,

Thank you very much for submitting your manuscript "Statistical prediction of microbial metabolic traits from genomes" for consideration at PLOS Computational Biology. As with all papers reviewed by the journal, your manuscript was reviewed by members of the editorial board and by several independent reviewers. The reviewers appreciated the attention to an important topic. Based on the reviews, we are likely to accept this manuscript for publication, providing that you modify the manuscript according to the review recommendations.

Sincerely,

Christos A. Ouzounis

Academic Editor

PLOS Computational Biology

Zhaolei Zhang

Section Editor

PLOS Computational Biology

Reviewer's Responses to Questions

**Comments to the Authors:**

Reviewer #1: This manuscript by Li et al discusses various methods to predict metabolic function using genomes. The main findings are that phylogenetic predictions are accurate provided that you have a close match in your database and that very large datasets can enable stronger, more mechanistic, and phylogeny-independent predictions. The writing is clear and easy to follow, and the conclusions are generally well supported by the data. The paper is timely, as available phenotypic and genomic data sets are poised to grow, increasing the usability of machine learning methods for phenotypic predictions.

The novelty of this work is murky, as the main findings are generally well known to science. The phylogenetic conservation of metabolism is well known; clinical microbiologists have been using metabolism to taxonomically label organisms for many decades. It is also generally well appreciated that larger data sets are required for accurate machine feature lists are long, and that without sufficiently large data overfitting (e.g. learning the wrong genes) can be a problem. Regardless, the unifying framework in which these lessons are discussed and the clarity with which they are discussed may prove of value to the large group of researchers getting started on machine learning or microbial predictions from genomes.

Suggestions for improvement:

-More discussion of the well-known relationship between phylogeny and metabolism (e.g. use in the clinic) would be appreciated in the results and/or discussion. Right now it is primarily discussed as why out-of-clade fails – as almost a negative prior on phylogeny being informative, overplaying the novelty of this work.

-The authors note 14 carbon sources for which the RF classifier does well out-of-clade when using gene/presence absence but for which a nearest neighbor approach on the same underlying data does not do well. Are these 14 carbons sources the most likely to vary *within clades*, forcing the RF model to learn something independent of phylogeny? This might help us understand the kinds of datasets that are needed—for example, data sets that include deep sampling within clades might be particularly useful.

--A graph that compares all various discussed methods side by side might be particularly useful. It is hard to graphically understand how various methods compare to one another.

Small notes:

-Gralka et al report more strains and carbon sources than listed in this paper. Is there a reason for this?

-Page 11: It seems that the KO presence absence is actually slightly better than 16S alone (84 vs 76; 24 vs 13)? Why do the authors downplay this?

-Page 8, top: After discussion of very well thought out metrics, it is not discussed how the just discussed classifier does on those metrics (Its in the figure but missing from the text).

-Page 8, bottom: Figure 5 referenced, I believe this is Figure S5.

-Page 11: For the sentence beginning with “Training nearest-neighbor classifiers using KO…” is not clear what the numbers in parentheses represent as comparators too, as no small numbers akin to 24/100 are presented earlier in the paragraph.

-Page 18, second paragraph: predicts->predict

-Page 19: It is not clear to me if rnfG is being used to predict phylogeny, perhaps I am missing something.

-Page 20: Top paragraph, extra “used” in “studied used”

-Page 24: The sentence “In particular, we show..” should have a qualification that says “some carbon sources”

-Page 24: Not clear for a naiive reader what “respect the phylogenetic structure” means, perhaps a better verb would be more clear.

Reviewer #2: Finding genomic signatures of microbial metabolic activity from communities is extremely challenging and directly linking microbial traits to environment and host health is the holy grail of microbial research. This paper successfully attempts to predict microbial traits from their genomic content from culture isolates and existing sequencing datasets – a clever combination of approaches with novel statistical tools.

The authors report a combinations of study approaches that go from simple to complex, leveraging robust statistical tools. They also use multiple datasets to test and verify the predictions. While I am not an expert in the mathematical aspects of the paper, I can vouch for the microbial mechanistic underpinnings of the approach. The paper uses strong theory and is based on clear microbial metabolic mechanisms. The focus on fundamental metabolic pathways makes it highly applicable to different sub-disciplines of microbiology.

The paper investigates the statistical links of genes to traits, sometimes using phylogenetic information to make those linkages, but what is more powerful is the attempt to look for the underlying mechanisms that examines the reason behind that statistical linkage. For example, the exploring of carbon utilization gene presence/absence patterns in the marine dataset to predict traits and how phylogenetic information is exploited to make predictions. The authors also present multiple examples of specific substrates or phylogenetic subsets to explain the mechanistic underpinning of their statistical predictions which makes the paper easy to read and understand. I also particularly liked the use of denitrification to test the random forest model and authors suggestion that the model does not necessarily detect mechanistically linked genes directly responsible for the trait but rather other genes that many be correlated with it. Indeed, in some cases the linkages are not linked to exact genes responsible for the traits and the mechanism of interest. But as the authors themselves suggest sometimes its ok to simply detect a gene that can be used to predict a process without knowing the underlying mechanism of the link of the gene to the process.

Authors discuss the results very well and provide some suggestions and recommendations for future work. This work will certainly be useful in guiding experiments and modelling endeavours to upscale from single cells to complex communities.

Methods are provided in detail, the data used in the paper are publicly available so is the data analysis code. I haven’t explored this in detail but skimming through I get the impression that it is well structured and easy to access.

It is quite a rare case that I am not making any suggestions to improve the paper. It is quite a polished manuscript ready for publication.

**Have the authors made all data and (if applicable) computational code underlying the findings in their manuscript fully available?**

Reviewer #1: Yes

Reviewer #2: Yes

PLOS authors have the option to publish the peer review history of their article (what does this mean?). If published, this will include your full peer review and any attached files.

Reviewer #1: No

Reviewer #2: No

Figure Files:

Data Requirements:

Reproducibility:

References:

---

## [Decision Letter · Decision Letter 1]

8 Nov 2023

Dear Dr. Kuehn,

Thank you very much for submitting your manuscript "Statistical prediction of microbial metabolic traits from genomes" for consideration at PLOS Computational Biology. As with all papers reviewed by the journal, your manuscript was reviewed by members of the editorial board and by several independent reviewers. The reviewers appreciated the attention to an important topic. Based on the reviews, we are likely to accept this manuscript for publication, providing that you modify the manuscript according to the review recommendations.

Sincerely,

Christos A. Ouzounis

Academic Editor

PLOS Computational Biology

Zhaolei Zhang

Section Editor

PLOS Computational Biology

Reviewer's Responses to Questions

**Comments to the Authors:**

Reviewer #1: I feel obligated to push back on the manner in which the prior expectation of correlation with phylogeny was discussed. The manuscript now includes these sentences: "It is important to recognize that the correlation between phylogeny and metabolic traits is known and has been recognized clinically and in the lab as an approach to identifying taxa from metabolic profiling [29,30]. However, our results provide a quantitative route to making such predictions, potentially providing new insights in other applications such as the human gut microbiome, where large libraries of isolates are available [31]. "

The first thing I have is that this statement give an incorrect impression. This is not the first paper to make a quantitative connection between phylogeny and metabolic traits. Here are some examples:

https://doi.org/10.1038/ismej.2012.160

https://doi.org/10.1038/nbt.2676

[These are just two such papers in a broad sea of literature]

The second thing is that this nod to the literature is coming rather late in the paper, whereas one would expect it in the introduction.

Reviewer #2: I have no further comments.

**Have the authors made all data and (if applicable) computational code underlying the findings in their manuscript fully available?**

Reviewer #1: Yes

Reviewer #2: Yes

PLOS authors have the option to publish the peer review history of their article (what does this mean?). If published, this will include your full peer review and any attached files.

Reviewer #1: No

Reviewer #2: No

Figure Files:

Data Requirements:

Reproducibility:

References:

---

## [Decision Letter · Decision Letter 2]

22 Nov 2023

Dear Dr. Kuehn,

We are pleased to inform you that your manuscript 'Statistical prediction of microbial metabolic traits from genomes' has been provisionally accepted for publication in PLOS Computational Biology.

Best regards,

Christos A. Ouzounis

Academic Editor

PLOS Computational Biology

Zhaolei Zhang

Section Editor

PLOS Computational Biology

Reviewer's Responses to Questions

**Comments to the Authors:**

Reviewer #1: The manuscript is improved by the inclusion of the new changes, thanks for making them.

I encourage the authors to add a bit more narrative to the readme page on the github to help a new reader get oriented, the github and data repository are otherwise easy to navigate.

**Have the authors made all data and (if applicable) computational code underlying the findings in their manuscript fully available?**

Reviewer #1: Yes

PLOS authors have the option to publish the peer review history of their article (what does this mean?). If published, this will include your full peer review and any attached files.

Reviewer #1: No

---

## [Editor Report · Acceptance letter]

4 Dec 2023

PCOMPBIOL-D-23-01320R2 

Statistical prediction of microbial metabolic traits from genomes

Dear Dr Kuehn,

I am pleased to inform you that your manuscript has been formally accepted for publication in PLOS Computational Biology. Your manuscript is now with our production department and you will be notified of the publication date in due course.

With kind regards,

Anita Estes
